# WavNAF: Learning Wave Propagation Priors for Neural Acoustic Fields

## Abstract

Room acoustics modeling requires capturing intricate wave phenomena such as reflections, refractions, and diffractions beyond direct sound propagation. Recent neural acoustic synthesis methods have improved acoustic realism but typically focus only on straight sound paths and coarse reverberation, missing detailed interactions like diffraction or multi-order reflections. We propose WavNAF, a neural framework that leverages physically-informed wave propagation priors to explicitly capture complex acoustic interactions. We generate these priors by numerically solving the wave equation with the Finite-Difference Time-Domain (FDTD) method, which directly simulates wave-based acoustic behavior that geometric methods cannot capture. Specifically, we extract essential acoustic parameters for FDTD, such as wave speed and density, from visual scene geometry encoded by Neural Radiance Fields (NeRF). We then generate physically-informed pressure maps and encode them via a feature extractor to learn wave propagation priors that capture intricate acoustic phenomena. To address the inherent computational cost issue of FDTD, we introduce a novel Neural Acoustic Scaling Module, inspired by traditional acoustic scale model. This module adaptively recalibrates encoded pressure map features from temporally compressed simulations to efficiently estimate accurate full-scale Room Impulse Responses. Experimental results demonstrate that WavNAF achieves substantial improvements in acoustic quality across various evaluation metrics compared to existing state-of-the-art methods.

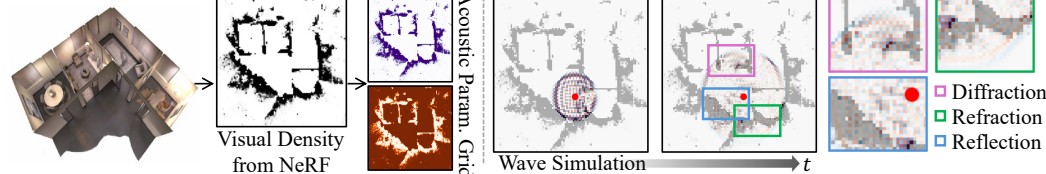

Figure 1: Our WavNAF framework uses Neural Radiance Fields (NeRF) to extract a normalized visual density grid encoding scene geometry, from which acoustic parameters, including sound speed and acoustic density, are directly derived. This enables wave-equation-based simulations that inherently capture complex acoustic phenomena, such as diffraction, refraction, and reflection, without explicit geometric modeling.

## 1 Introduction

Room acoustics modeling requires capturing intricate wave phenomena beyond direct sound propagation, including reflections, refractions, and diffractions. Traditional geometric methods (Krokstad et al., 1968; Cremer & Müller, 1948) efficiently handle direct paths but are inherently unable to model wave phenomena like diffraction due to their ray-based formulation. Recent neural approaches (Luo et al., 2022; Brunetto et al., 2024; Bhosale et al., 2024) have improved generalization and efficiency but remain limited to approximating coarse reverberation without capturing detailed wave physics.

We propose WavNAF, a framework that bridges physics-based simulation and neural acoustic modeling by leveraging wave propagation priors from Finite-Difference Time-Domain (FDTD) simulations (Botteldooren, 1995). FDTD numerically solves the wave equation on spatial grids, naturally capturing diffraction, interference, and multi-order reflections through physics-governed update rules.

However, integrating FDTD into neural acoustic modeling presents three fundamental challenges: deriving scene geometry from visual data, effectively encoding wave physics for neural learning, and managing computational bottlenecks under stringent stability constraints.

First, we derive essential acoustic parameters from visual scene geometry captured via NeRF (Mildenhall et al., 2020). We extract volumetric density fields and transform them into 2D acoustic parameter grids through vertical max-pooling, mapping visual opacity to spatially-varying sound speed and acoustic density. This enables physics-informed wave simulations without explicit material annotations, generating pressure maps that inherently encode complex wave phenomena.

Second, we effectively integrate FDTD-simulated pressure maps as robust wave propagation priors into neural acoustic fields. Unlike previous methods that rely on geometric features or visual cues, our approach directly learns from pressure maps through a dedicated feature extractor. By processing these pressure maps that inherently capture diffraction, interference, and multi-order reflections without requiring scene geometry, we demonstrate that the physics-governed wave patterns provide strong inductive bias for acoustic field learning, achieving superior performance even with simplified material parameters.

Third, we introduce a Neural Acoustic Scaling Module to alleviate computational bottlenecks inherent in standard FDTD methods. Inspired by traditional acoustic scale models (Suzuki & Hidaka, 2019; Barron, 2002; Rindel, 2011) where early reflections scale well but late reverberation deviates systematically, our module learns adaptive transformations from temporally compressed simulations to full-scale impulse responses, ensuring efficient and accurate representation while maintaining acoustic fidelity.

## 2 RELATED WORK

**Implicit Neural Representations** Early neural methods for acoustic synthesis primarily utilized implicit representations to model room acoustics. Neural Acoustic Fields (NAF) (Luo et al., 2022) directly mapped emitter-listener positions to impulse responses, enabling continuous spatial interpolation but providing limited geometry awareness. INRAS (Su et al., 2022) enhanced implicit modeling by explicitly sampling bounce points from scene meshes, enabling geometry-aware inference of reflections and reverberation. However, both methods typically have limitations to capture intricate wave phenomena such as diffraction and complex reverberation patterns accurately.

**Image-guided Acoustic Modeling** Visual information has been effectively utilized to condition acoustic synthesis models. AV-NeRF (Liang et al., 2023a) and Neural Acoustic Context Field (NACF) (Liang et al., 2023b) represent prominent examples employing intermediate visual renderings to inform acoustic predictions. Additionally, NACF explicitly leverages bounce points sampled from 3D scene meshes, similar to INRAS (Su et al., 2022), to achieve geometry-aware acoustic modeling. These approaches have significantly improved acoustic realism by integrating visual scene context. Nevertheless, these visual rendering-based methods typically exhibit limitations in fully capturing intricate wave interactions, particularly those involving acoustic interactions with scene elements beyond rendered viewpoints.

**Explicit Spatial Acoustic Representations** Recent work such as AV-GS (Bhosale et al., 2024) and NeRAF (Brunetto et al., 2024) leverage explicit 3D scene representations derived from advanced visual modeling techniques to enrich acoustic modeling. AV-GS employs a Gaussian Splatting-based (Kerbl et al., 2023) representation, using 3D Gaussian points augmented with audio-guidance parameters to explicitly encode geometry and material characteristics. In contrast, NeRAF utilizes volumetric features directly extracted from a NeRF (Mildenhall et al., 2020), capturing geometry and appearance cues as voxel grids. These approaches encourage neural models to more explicitly leverage local geometric information. However, they still lack explicit modeling of interactions between scene geometry and acoustic wave propagation. Consequently, accurately capturing complex wave phenomena remains challenging.

**Ray-casting and Acoustic Volume Rendering** Ray-based approaches like Scene Occlusion-Aware Acoustic Fields (SOAF) (Gao et al., 2024) and Acoustic Volume Rendering (AVR) (Lan et al., 2024) have enhanced acoustic realism by modeling acoustic occlusion and spatial variations within continuous volumetric frameworks. These methods incorporate distance-based acoustic energy fields and frequency-dependent transmittance characteristics to improve realism. Despite these advances,

there remains limitations in accurately modeling complex wave phenomena such as diffraction around obstacles or interference patterns caused by multiple wavefront interactions, highlighting the advantage of using wave-equation-based approaches that naturally capture these acoustic behaviors.

## 3 BACKGROUND

### 3.1 WAVE SIMULATION WITH FINITE-DIFFERENCE TIME-DOMAIN (FDTD)

Acoustic wave propagation is formulated by the following wave equation (Kowalczyk & van Walstijn, 2009):

$$\frac{\partial^2 p}{\partial t^2} = c^2 \nabla^2 p + f(X, t), \tag{1}$$

where $p(X, t)$ is acoustic pressure at spatial position $X = (x, y, z)$ and time $t$, $c$ is the speed of sound, and $f(X, t)$ is the acoustic source term.

**Discretization and Computational Scheme** For computational efficiency, the above acoustic wave equation can be solved in two dimensions using FDTD schemes (Hamilton, 2016). The simulation domain is discretized into a spatial grid with indices $(i, j)$ along the $x$ and $y$ axes, and time is discretized into steps indexed by $n$. As a result, the simulation iteratively updates pressure $p^n$ and velocity $v^n = (v_x^n, v_y^n)$ for each time step $n$, as follows (detailed derivation process in Appendix B.4):

$$\text{Pressure Gradient:} \quad \frac{\partial p}{\partial x} \approx \frac{p_{i+1,j} - p_{i,j}}{\Delta x}, \quad \frac{\partial p}{\partial y} \approx \frac{p_{i,j+1} - p_{i,j}}{\Delta y}, \tag{2}$$

$$\text{Velocity Update:} \quad v_x^{n+1} = v_x^n - \frac{\Delta t}{\rho_{i,j} \Delta x} \frac{\partial p}{\partial x}, \quad v_y^{n+1} = v_y^n - \frac{\Delta t}{\rho_{i,j} \Delta y} \frac{\partial p}{\partial y}, \tag{3}$$

$$\text{Velocity Divergence:} \quad \nabla \cdot v \approx \frac{v_{x,i,j} - v_{x,i-1,j}}{\Delta x} + \frac{v_{y,i,j} - v_{y,i,j-1}}{\Delta y}, \tag{4}$$

$$\text{Pressure Update:} \quad p^{n+1} = p^n - \rho_{i,j} c_{i,j}^2 \Delta t (\nabla \cdot v), \tag{5}$$

where terms $\rho_{i,j}$ and $c_{i,j}$ denotes acoustic density and speed of sound at the $(i, j)$-th grid cell. By iteratively applying these update rules, the FDTD method naturally captures complex wave phenomena including diffraction, interference, and scattering that arise from the underlying physics of the wave equation, without requiring explicit modeling of each individual effect.

**CFL Condition for Stability** To ensure numerical stability and convergence, the Courant-Friedrichs-Lewy (CFL) condition (Courant et al., 1928) must be satisfied. This condition constrains the time step $\Delta t$ relative to spatial discretization $\Delta x$ and $\Delta y$, preventing numerical instabilities and maintaining physically accurate wave propagation. For anisotropic grids, where $\Delta x \neq \Delta y$, the CFL condition is:

$$\Delta t \leq \frac{\Delta x \Delta y}{c\sqrt{\Delta x^2 + \Delta y^2}}. \tag{6}$$

We choose the simulation step size as the maximum permissible value under this CFL condition, denoted as $\Delta t_{\text{CFL}}$.

### 3.2 ACOUSTIC SCALE MODEL

The acoustic wave equation exhibits theoretical scale invariance under proportional scaling of spatial and temporal dimensions. Specifically, consider scaling spatial and temporal variables by a factor $\lambda$, and define the scaled pressure field $p'(X, t)$ as follows:

$$p'(X, t) := p(X/\lambda, t/\lambda). \tag{7}$$

Under this definition, spatial and temporal derivatives transform as:

$$\frac{\partial^2 p'}{\partial t'^2} = \frac{1}{\lambda^2} \frac{\partial^2 p}{\partial t^2}, \quad \nabla^2 p' = \frac{1}{\lambda^2} \nabla^2 p. \tag{8}$$

Substituting these relationships into the wave equation, we obtain:

$$\frac{\partial^2 p'}{\partial t'^2} - c^2 \nabla^2 p' = \frac{1}{\lambda^2} \left( \frac{\partial^2 p}{\partial t^2} - c^2 \nabla^2 p \right) = 0. \tag{9}$$

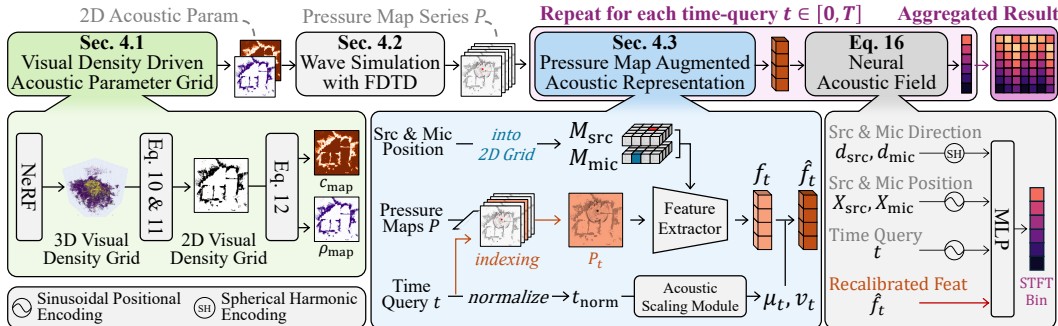

Figure 2: Overview of the WavNAF pipeline. A 3D voxel density grid is first generated via NeRF and converted into 2D acoustic parameter grids. Using these grids, we simulate acoustic pressure maps through FDTD wave simulation. For a given time query, the corresponding pressure map is combined with one-hot encoded source and microphone positions, then processed by a feature extractor and scaled by our neural acoustic scaling module. Finally, an MLP predicts the log-magnitude STFT of the room impulse response from these scaled features.

This demonstrates the theoretical invariance of the acoustic wave equation under simultaneous scaling of space and time. Such invariance motivates the use of acoustic scale models (Cremer & Müller, 1948; Beranek, 1992; Kuttruff & Vorländer, 2024), which are physically reduced versions of rooms or auditoriums designed to predict full-scale acoustic behavior by propotional scaling.

However, in practical scenarios, simple linear scaling fails to accurately predict full-scale acoustics due to frequency-dependent effects such as increased high-frequency air absorption, material differences, and measurement hardware limitations (Suzuki & Hidaka, 2019; Barron, 2002; Rindel, 2011). These practical constraints prevent straightforward linear scaling of acoustic responses, particularly affecting the accuracy of late reverberation. Therefore, accurately reconstructing full-scale acoustic responses from scaled-down simulations necessitates more sophisticated, non-linear acoustic scaling methods.

## 4 METHOD

We propose **WavNAF**, a framework that integrates physics-informed wave propagation priors into neural RIR estimation through FDTD simulations (Figure 2). WavNAF derives acoustic parameters from visual scene geometry (Section 4.1), performs FDTD simulations to generate pressure maps capturing complex wave phenomena (Section 4.2), and employs a Neural Acoustic Scaling Module to transform compressed simulations into full-scale RIRs (Section 4.3). This approach enables high-fidelity acoustic modeling while maintaining computational efficiency.

### 4.1 ACOUSTIC PARAMETER GRID

To perform acoustic wave simulations with the FDTD method, each grid cell must define essential acoustic parameters describing local sound speeds and densities. However, these acoustic parameters are generally not directly available. To address this, we leverage NeRF to obtain spatially varying acoustic density information from visual data.

**Neural Radiance Field** The scene is represented by an $N^3$ voxel grid, with coordinates normalized to the range [0, 1], following NeRF's scene-contraction method (Tancik et al., 2023). NeRF encodes scene structures through a learned volumetric density function $\sigma(X)$. Querying this visual density at discrete spatial coordinates produces a density grid, which is used to define local acoustic parameters within our FDTD simulation.

**Conversion from Visual Density Field to Acoustic Parameters** The visual density values $\sigma(x, y, z)$ are first converted into alpha compositing values $\alpha_{3D}$:

$$\alpha_{3D}(x, y, z) = 1 - \exp(-\sigma(x, y, z) \cdot \delta), \tag{10}$$

where $\delta$ is a small positive constant. To obtain a 2D alpha map at microphone height $z_{\mathrm{mic}}$, we apply a maximum pooling operation over a small vertical range $\Delta z$ around the targeted $z$-coordinates:

$$\alpha_{\mathrm{2D}}(x, y) = \max_{z \in [z_{\mathrm{mic}} - \Delta z, \, z_{\mathrm{mic}} + \Delta z]} \alpha_{\mathrm{3D}}(x, y, z). \tag{11}$$

This vertical pooling procedure effectively smooths noisy surface details and implicitly captures geometric influences along the $z$-axis. The resulting 2D alpha map $\alpha_{\mathrm{2D}} \in \mathbb{R}^{N \times N}$ is then mapped onto acoustic parameters essential for FDTD simulations:

$$c_{\mathrm{map}}(x, y) = c_{\min} + (c_{\max} - c_{\min})(1 - \alpha_{\mathrm{2D}}(x, y)), \quad \rho_{\mathrm{map}}(x, y) = \rho_{\mathrm{air}}(1 + \alpha_{\mathrm{2D}}(x, y)), \tag{12}$$

where $c_{\max}$ is set to typical sound speed 343 m/s, and $c_{\min}$ is empirically chosen as $0.9 \cdot c_{\max}$. The parameter $\rho_{\mathrm{air}}$ represents the air density and set to 1.21 kg/m$^3$. While these mappings use empirically chosen parameters, our method demonstrates robust performance across various parameter settings. This robustness demonstrates that the physics-based update rules provide effective inductive bias for learning wave propagation patterns, even when essential physical parameters are approximated. Our physics-informed approach provides valuable neural guidance regardless of specific parameter choices, consistently yielding superior results compared to baseline methods across a wide range of $c_{\min}/c_{\max}$ ratios (detailed ablation studies in Appendix D).

## 4.2 FDTD Wave Simulation

Based on the acoustic parameter grids from Section 4.1, we simulate wave propagation using the Finite-Difference Time-Domain (FDTD) method. This simulation scheme iteratively updates the pressure and velocity fields to capture complex acoustic behaviors, as detailed in Algorithm 1. The spatial grid resolution is set to $N \times N$.

We adopt a scene normalization process based on NeRF's scene contraction (Tancik et al., 2023). This procedure maps the 3D scene geometry into a normalized coordinate space in the range $[0, 1]^3$, allowing the simulation to operate independently of the original room size or aspect ratio.

To determine the physical grid cell sizes $\Delta x, \Delta y$, we first empirically align audio sensor positions (i.e., source and microphone) to the normalized scene geometry. This alignment yields a normalization factor representing the full extent of the simulation domain in each spatial axis. We then compute $\Delta x$ and $\Delta y$ by dividing these spatial extents by the grid resolution $N$.

While the acoustic wave equation is theoretically scale-invariant under proportional spatial-temporal scaling, directly simulating full-scale wave behavior is computationally expensive. Therefore, we adopt a normalized, scaled-down simulation space that preserves physical fidelity through this scale-invariance. This allows for more efficient simulation while maintaining acoustic fidelity.

To mitigate boundary artifacts and emulate realistic wave energy dissipation, we introduce two damping mechanisms. First, a sponge layer (Cerjan et al., 1985) applies spatially varying damping near the simulation boundaries to suppress spurious wave reflections. The damping coefficient $\gamma_{\mathrm{sponge}}(i, j)$ transitions smoothly from a minimum 0.9 at the boundary to 1.0 toward the interior. Second, a global damping factor $\gamma_{\mathrm{global}} = 0.99$ is uniformly applied across the domain to model natural air attenuation. These two damping mechanisms are combined at each timestep as follows:

$$p_{i,j}^{(n)} \leftarrow p_{i,j}^{(n)} \cdot \gamma_{\mathrm{sponge}}(i, j) \cdot \gamma_{\mathrm{global}}, \quad v_{i,j}^{(n)} \leftarrow v_{i,j}^{(n)} \cdot \gamma_{\mathrm{sponge}}(i, j) \cdot \gamma_{\mathrm{global}}. \tag{13}$$

The resulting sequence of pressure maps $P \in \mathbb{R}^{T \times N \times N}$ explicitly encodes wave-based acoustic interactions and serves as a physically grounded prior for the subsequent neural acoustic field model.

## 4.3 Neural Acoustic Field

We leverage pressure maps from wave simulations as physically-informed priors for neural RIR estimation, enabling the model to learn representations that capture intricate wave phenomena.

While FDTD effectively captures wave propagation, full-scale RIR simulation requires 25× more steps overhead due to CFL constraints. Traditional acoustic scale modeling (Suzuki & Hidaka, 2019; Barron, 2002; Rindel, 2011) shows early reflections scale well but late reverberation deviates systematically.

We introduce a neural acoustic scaling module that learns time-dependent transformations from compressed simulations to full-scale responses. This adaptive approach bridges the gap between

---

**Algorithm 1:** Acoustic Wave Simulation with FDTD

---

**Data:** Acoustic parameter grids $c_{\text{map}} \in \mathbb{R}^{N \times N}, \rho_{\text{map}} \in \mathbb{R}^{N \times N}, \gamma_{\text{sponge}} \in \mathbb{R}^{N \times N}$; Normalized source positions $(x_{\text{src}}, y_{\text{src}}) \in [0,1]^2$; Simulation parameters: grid cell size $\Delta x, \Delta y$, time step size $\Delta t$, initial source amplitude $A_{\text{init}}$, global damping coefficient $\gamma_{\text{global}}$, number of frames $T$, step interval $m$ (default: 1)

**Result:** Series of pressure maps $P[T, N, N]$

Initialize pressure field $p^{(0)} \in \mathbb{R}^{N \times N}$, velocity fields $v_x^{(0)}, v_y^{(0)} \in \mathbb{R}^{N \times N} \leftarrow 0$;

Convert continuous source positions to discretized grid indices:;

$i \leftarrow \text{round}\,(x_{\text{src}} \times (N-1)), j \leftarrow \text{round}\,(y_{\text{src}} \times (N-1))$;

Set initial source amplitude: $p^{(0)}[i,j] \leftarrow A_{\text{init}}$;

total_steps $\leftarrow T \times m$;

Initialize pressure map series $P[T, N, N] \leftarrow 0$;

frame_idx $\leftarrow 0$;

**for** $n = 1$ **to** total_steps **do**

    $\partial_x p, \partial_y p \leftarrow \text{Gradient}\,(p^{(n-1)}, \Delta x, \Delta y)$ (Eq. 2);

    $v_x^{(n)}, v_y^{(n)} \leftarrow \text{UpdateVelocity}\,(v_x^{(n-1)}, v_y^{(n-1)}, \partial_x p, \partial_y p, \rho_{\text{map}}, \Delta t, \Delta x, \Delta y)$ (Eq. 3);

    $\nabla \cdot v^{(n)} \leftarrow \text{Divergence}\,(v_x^{(n)}, v_y^{(n)}, \Delta x, \Delta y)$ (Eq. 4);

    $p^{(n)} \leftarrow \text{UpdatePressure}\,(p^{(n-1)}, \nabla \cdot v^{(n)}, c_{\text{map}}, \rho_{\text{map}}, \Delta t)$ (Eq. 5);

    $p^{(n)}, v_x^{(n)}, v_y^{(n)} \leftarrow \text{ApplyDamping}\,(p^{(n)}, v_x^{(n)}, v_y^{(n)}, \gamma_{\text{sponge}}, \gamma_{\text{global}})$ (Eq. 13);

    **if** $n \bmod m = 0$ **then**

        $P[\text{frame\_idx}] \leftarrow p^{(n)}$;

        frame_idx $\leftarrow$ frame_idx $+ 1$;

    **end**

**end**

---

simulation intervals $\Delta t_{\text{CFL}}$ and target STFT intervals $\Delta t_{\text{STFT}}$, enabling efficient and accurate RIR reconstruction.

**Wave Feature from Pressure Map for Learning Wave Propagation Prior** Given a time query $t \in [0, T]$, we select the pressure map $P_t \in \mathbb{R}^{N \times N}$ from generated pressure map series $P$. Additionally, one-hot encoded positional maps $M_{\text{src}}, M_{\text{mic}} \in \mathbb{R}^{N \times N}$ for source and microphone positions are concatenated as follows:

$$I_t = [P_t, M_{\text{src}}, M_{\text{mic}}]. \tag{14}$$

Crucially, the feature extraction network $F_\theta$ processes this input to learn representations $f_t = F_\theta(I_t)$ that serve as wave propagation priors, derived purely from pressure patterns without any explicit scene geometry information. The network learns these acoustic priors solely from the pressure map that inherently encodes wave phenomena such as diffraction, interference, and multi-order reflections. This design choice is intentional: we aim to demonstrate that wave physics itself, as captured in pressure distributions, provides valuable priors for acoustic field learning beyond simple geometric structure.

**Neural Acoustic Scaling Module for Temporal Alignment** Due to the CFL stability condition, the FDTD time step $\Delta t_{\text{CFL}}$ is much finer than the target STFT hop size $\Delta t_{\text{STFT}}$, which creates a systematic temporal mismatch between simulated and full-scale RIRs, especially in the late-reverberation region where cumulative energy-decay errors are amplified. To compensate for this, we introduce an adaptive neural scaling module that learns time-dependent transformations. Given a normalized time query $t_{\text{norm}} \in [0, 1]$, it recalibrates the extracted simulation features $f_t$ as

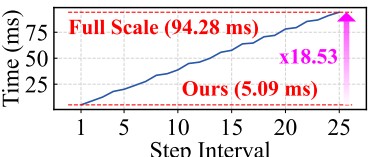

Figure 3: Wave simulation computational cost per step interval.

$$\hat{f}_t = \mu_t \odot f_t + \nu_t, \tag{15}$$

where $\mu_t = \text{MLP}_\mu(t_{\text{norm}})$ and $\nu_t = \text{MLP}_\nu(t_{\text{norm}})$, and $\odot$ denotes element-wise multiplication. This time-varying affine transformation applies minimal correction to early reflections while allowing

larger adjustment in the late tail, effectively aligning the temporal behavior of simulated features with the target RIR representation and bridging the gap between $\Delta t_{\text{CFL}}$ and $\Delta t_{\text{STFT}}$ (see Appendix E.2).

**Neural Impulse Response Estimation** We predict impulse responses by directly estimating their STFTs with a unified feature-fusion MLP.

$$\text{NAF}(\hat{f}_t, X_{\text{mic}}, d_{\text{mic}}, X_{\text{src}}, d_{\text{src}}, t_{\text{norm}}) \rightarrow \hat{s}_t \in \mathbb{R}^{C \times F}, \tag{16}$$

where $\hat{s}_t$ represents the STFT magnitude spectrum at time $t$. Given microphone and source positions $X_{\text{mic}}, X_{\text{src}}$ and orientations $d_{\text{mic}}, d_{\text{src}}$, we apply sinusoidal positional encoding for position $X$ and time $t_{\text{norm}}$, spherical harmonic encoding for direction $d$. Then, all elements including the pressure map feature $\hat{f}_t$ are concatenated into a single vector. This vector is processed by a feature-fusion MLP architecture—similar to previous work (Brunetto et al., 2024)—to predict the single STFT time bin $\hat{s}_t$. By iterating over all $T$ queries, we assemble the full STFT $\hat{S} \in \mathbb{R}^{C \times F \times T}$. Finally, the Griffin-Lim algorithm (Perraudin et al., 2013; Griffin & Lim, 1984) reconstructs the full-scale waveform RIR form $\hat{S}$.

### 4.4 LEARNING OBJECTIVE

Following prior works (Liang et al., 2023a; Brunetto et al., 2024), we train the visual model for sufficient iterations to establish a stable geometric encoding before incorporating acoustic model training. The loss function of visual model $\mathcal{L}_{\text{V}}$ is the same as original NeRF (Mildenhall et al., 2020):

$$\mathcal{L}_{\text{V}} = ||C(r) - \hat{C}(r)||_2^2, \tag{17}$$

where $C(r)$ is the ground-truth pixel color and $\hat{C}(r)$ is the rendered color at the ray $r$. Following the previous work (Brunetto et al., 2024), we use a combination of spectral loss $\mathcal{L}_{\text{SL}}$ (Défossez et al., 2018a) and spectral convergence loss $\mathcal{L}_{\text{SC}}$ (Arık et al., 2018) as our audio loss $\mathcal{L}_{\text{A}}$:

$$\mathcal{L}_{\text{SL}} = ||\log(|S| + \epsilon) - \log(|\hat{S}| + \epsilon)||_2^2, \quad \mathcal{L}_{\text{SC}} = \frac{|||S| - |\hat{S}|||_F}{|||S|||_F}, \quad \mathcal{L}_{\text{A}} = \lambda_{\text{SL}}\mathcal{L}_{\text{SL}} + \lambda_{\text{SC}}\mathcal{L}_{\text{SC}}, \tag{18}$$

where $\hat{S}$ is the predicted STFT, $S$ is the ground-truth STFT, $||.||_F$ denotes the Frobenius norm, $||.||_2$ denotes L2 norm and $\epsilon = 10^{-3}$. The final learning objective $\mathcal{L}$ can be formulated as:

$$\mathcal{L} = \mathcal{L}_{\text{V}} + \lambda_{\text{A}}\mathcal{L}_{\text{A}}. \tag{19}$$

## 5 EXPERIMENTS

### 5.1 EXPERIMENTAL SETTINGS

#### 5.1.1 DATASET

Following prior works (Su et al., 2022; Brunetto et al., 2024; Bhosale et al., 2024), we compare our method against baselines on top of representative benchmarks specified below.

**SoundSpaces** (Chen et al., 2020) is a synthetic 3D acoustic simulator that provides stereo RIR (for discrete head orientations: 0, 90, 180, and 270) at the receiver positions of a 2D spatial grid. We follow our baselines (Luo et al., 2022; Su et al., 2022; Brunetto et al., 2024; Liang et al., 2023a; Bhosale et al., 2024) and employ 6 selected (Straub et al., 2019) indoor scenes with varying degrees of complexity as: 2 scenes with a single rectangular room; 2 scenes with a single non-rectangular room; and 2 scenes with a complex layout with multiple rooms. The train-validation splits are following the conventions of prior works (Liang et al., 2023a; Bhosale et al., 2024; Brunetto et al., 2024).

**Real Acoustic Fields (RAF)** (Chen et al., 2024) is a multi-modal real-world 3D RIR dataset. It consists of precise 6DoF pose tracking data paired with high-quality RIR annotations and dense multi-view images. Acoustic data are collected via a pair of a loudspeaker and a microphone recording system (earful tower) placed at different locations for RIR estimation, and visual data are captured with a moving camera rig (eyeful tower) for multi-view image synthesis.

#### 5.1.2 EVALUATION METRIC

We evaluate each method using three standard acoustic metrics: **reverberation time (T60)** for late reverberation, **clarity (C50)** for acoustic clarity, and **early decay time (EDT)** for early reflections.

Table 1: **Quantitative comparison with state-of-the-art methods on SoundSpaces (left) and RAF (right) datasets.** Lower values of T60, C50, EDT, and STFT error (↓) indicate higher RIR quality. Checkmarks (✓) in the GT column denote methods using ground-truth mesh information.

| SoundSpaces | | | | | RAF | | | | | |
|---|---|---|---|---|---|---|---|---|---|---|
| Methods | GT | T60 ↓ | C50 ↓ | EDT ↓ | Methods | GT | T60 ↓ | C50 ↓ | EDT ↓ | STFT error ↓ |
| Opus-nearest | | 10.10 | 3.58 | 0.115 | Opus-nearest | | 10.03 | 0.76 | 0.021 | 0.49 |
| Opus-linear | | 8.64 | 3.13 | 0.097 | Opus-linear | | 10.19 | 0.86 | 0.029 | 0.92 |
| AAC-nearest | | 9.35 | 1.67 | 0.059 | AAC-nearest | | 22.83 | 1.97 | 0.064 | 1.04 |
| AAC-linear | | 7.88 | 1.68 | 0.057 | AAC-linear | | 25.64 | 2.49 | 0.085 | 1.26 |
| INRAS | ✓ | 3.14 | 0.60 | 0.019 | INRAS | ✓ | 8.01 | 0.79 | 0.025 | **0.36** |
| NACF | ✓ | 2.36 | 0.50 | **0.014** | NACF | ✓ | **6.62** | 0.59 | **0.017** | 0.39 |
| NACF w/ T | ✓ | **2.17** | 0.49 | **0.014** | NACF w/ T | ✓ | 7.31 | 0.59 | 0.018 | 0.39 |
| NAF | | 3.18 | 1.06 | 0.031 | NAF | | 10.08 | 0.71 | 0.021 | 0.64 |
| AV-NeRF | | 2.47 | 0.57 | 0.016 | AV-NeRF | | 8.11 | 0.73 | 0.021 | 0.39 |
| NeRAF | | 2.14 | 0.38 | 0.010 | NeRAF | | 7.47 | 0.61 | 0.020 | 0.17 |
| **WavNAF** | | **1.95** | **0.33** | **0.009** | **WavNAF** | | **7.43** | **0.61** | **0.019** | **0.16** |

Specifically, T60 measures the time for the impulse response to decay by 60 dB, C50 quantifies the energy ratio between the first 50ms and the remaining impulse response, and EDT assesses the initial decay rate, closely aligning with human auditory perception. On the RAF dataset, we additionally measures the **STFT error** (Luo et al., 2022; Défossez et al., 2018b), defined as the absolute error between predicted and ground-truth log-magnitude STFTs.

## 5.2 IMPLEMENTATION DETAILS

Whole FDTD computations are optimized with CUDA kernels for GPU acceleration, ensuring efficient memory usage and parallel execution. We use Nerfacto (Tancik et al., 2023) as our NeRF backbone and adopt ResNet50 (He et al., 2016) as the feature extractor for pressure map features. The spatial resolution for acoustic parameter grids is set to $128 \times 128$. We set the step interval $m$ as 1, corresponding to the maximum permissible time step under the CFL condition $\Delta t_{\text{CFL}}$ detailed in Section 3. The source amplitude $A_{\text{init}}$ is initially set to 1.0 and subsequently adjusted to be inversely proportional to the vertical distance between the source and microphone positions. To minimize unnecessary reflections at simulation boundaries—outside of the room—we apply a sponge layer (Cerjan et al., 1985) characterized by a minimum damping coefficient $\gamma_{\text{min}} = 0.9$ and additionally employ a global damping coefficient $\gamma_{\text{global}} = 0.99$ to simulate realistic air attenuation effect. We set batch size $B = 1024$.

## 5.3 RESULTS

We compare WavNAF with traditional audio encoding methods (AAC (International Organization for Standardization, 2006), Opus (Xiph.Org Foundation, 2012)) and advanced neural acoustic synthesis models (Luo et al., 2022; Su et al., 2022; Liang et al., 2023a;b; Brunetto et al., 2024). Neural models are categorized into those utilizing ground-truth (GT) mesh information, such as INRAS and NACF, and those without it.

On SoundSpaces, WavNAF consistently outperforms all baseline methods across evaluation metrics, demonstrating superior acoustic realism.

On RAF, WavNAF improves T60, EDT, and STFT error compared to neural methods without GT mesh data. These results confirm that WavNAF effectively leverages physically-informed wave propagation priors for state-of-the-art acoustic synthesis.

## 5.4 ABLATION STUDY

**Effect of Wave Feature.** We analyze the impact of our key contributions: the pressure map feature as a wave propagation prior and the neural acoustic scaling module (Table 2). The baseline *No condition* includes only positional, directional, and temporal query information. The *3D grid feature* refers to the static scene structure feature previously introduced in NeRAF (Brunetto et al., 2024). Introducing our pressure map feature significantly enhances the RIR quality across all evaluation metrics. Furthermore, the neural acoustic scaling module provides additional improvements, demonstrating the efficacy of dynamically adapting simulation features to match full-scale acoustic conditions.

Table 2: **Ablation study on feature conditioning method.** Performance difference between feature conditioning methods at SoundSpaces apartment 1

| Methods | T60 (%)↓ | C50 (dB)↓ | EDT (sec)↓ |
|---|---|---|---|
| no condition | 2.776 | 0.590 | 0.0151 |
| 3D grid feature | 2.725 | 0.574 | 0.0149 |
| only wave feature | 2.406 | 0.467 | 0.0122 |
| +) acoustic scale module | **2.306** | **0.446** | **0.0119** |

Table 3: **Ablation study on $z$-axis max pooling.** Performance difference between $z$-axis max pooling parameter at SoundSpaces apartment 1

| $\Delta z$ | T60 (%)↓ | C50 (dB)↓ | EDT (sec)↓ |
|---|---|---|---|
| 0 | 2.352 | 0.470 | 0.0124 |
| 1 | 2.450 | 0.463 | 0.0121 |
| 2 | **2.306** | **0.446** | **0.0119** |
| 3 | 2.387 | 0.457 | 0.0121 |

**Effect of $z$-axis Max Pooling.** We investigate the influence of the vertical max pooling size $\Delta z$ (Table 3). A setting $\Delta z = 0$ corresponds to using a single slice at the exact query height without any pooling. The results clearly demonstrate that employing an appropriate pooling size along the $z$-axis significantly enhances RIR quality, highlighting the benefit of aggregating alpha grid vertically to better capture acoustic interactions and reduce surface-level noise.

**Simulation Length.** We examine how varying the simulation length through different step intervals affects performance. Specifically, a step interval of 1 indicates that we utilize every simulation step, corresponding to a temporal resolution of $\Delta t_{\text{CFL}}$. Similarly, a step interval of $m$ implies selecting every $m$-th pressure map, resulting in an effective temporal resolution of $m \cdot \Delta t_{\text{CFL}}$. The detailed simulation procedure for different step intervals is outlined in Algorithm 1. For the SoundSpaces apartment 1 scenario, a step interval of 25 corresponds to a full-length simulation. Due to the prohibitive computational cost associated with larger step intervals (longer simulation length), we limit our experiments to a maximum step interval of 3. Figure 4 compares the three main acoustic metrics across training iterations and simulation time for various step intervals. Increasing the step interval significantly raises the computational cost, by up to approximately 20 times (Figure 3). However, performance differences among these intervals remain minimal. These results indicate that our framework effectively learns the wave propagation prior, achieving comparable accuracy at substantially lower computational costs compared to conducting full-length simulations.

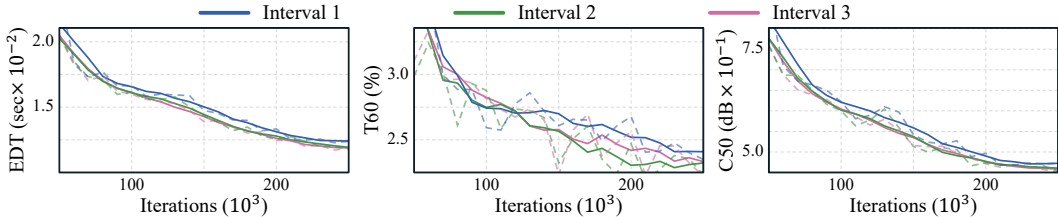

Figure 4: Comparison of metrics over iterations for different simulation lengths. This demonstrates that our method achieves robust results efficiently, even with temporally compressed simulations.

## 6 DISCUSSION

**Limitation** Despite the performance improvements achieved by our wave propagation prior learning strategy, there are several limitations. First, due to the computational demands, our current method utilizes only 2D wave simulations. Extending our framework to incorporate 3D simulations by including the vertical axis would likely yield additional performance gains. Second, WavNAF shows sensitivity to the alignment between visual and acoustic coordinate systems, and currently requires empirical tuning to ensure precise coordinate alignment. Future work should investigate systematic methods to mitigate these alignment sensitivities.

**Conclusion** In this paper, we presented WavNAF, a novel framework that effectively captures complex acoustic interactions by integrating FDTD wave simulations with neural acoustic fields. WavNAF leverages visual geometry extracted via NeRF to inform acoustic parameter grids, enabling modeling of intricate acoustic phenomena including reflection, refraction, and diffraction. Moreover, our neural acoustic scaling module significantly enhances computational efficiency, allowing accurate full-scale RIR estimation from temporally scaled simulations. Experimental results confirm that WavNAF achieves substantial improvements in acoustic quality, outperforming prior methods across multiple standard metrics. Future work includes expanding our framework to full 3D simulations and developing systematic methods to address visual-acoustic coordinate alignment sensitivities.

## 7 Reproducibility Statement

Experimental settings for both training and evaluation are described in Sec 5.2. Detailed hyperparameter settings and network configuration for each model variant are described in Appendix B.2.1 and Appendix B.1. And detailed FDTD simulation parameters are described in Appendix B.3. We plan to release the code for further reproducibility.

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

## A   TASK DEFINITION AND DATASET DETAILS

### A.1   TASK DEFINITION

The task of WavNAF, is to synthesize a high-fidelity Room Impulse Response (RIR) from a 3D scene represented by multi-view images. Given the visual information and arbitrary positional queries for a sound source and receiver, the model is trained to predict the corresponding RIR waveform that captures complex acoustic phenomena.

### A.2   REQUIRED INPUTS

The WavNAF model requires the following inputs to generate an RIR:

- **Multi-view RGB images with known camera poses:** A set of images capturing the 3D scene from various viewpoints, used for visual reconstruction via Neural Radiance Fields (NeRF).
- **Query source position** $(x_{\mathbf{src}}, y_{\mathbf{src}}, z_{\mathbf{src}})$**:** The 3D coordinates of the sound source.
- **Query receiver position** $(x_{\mathbf{mic}}, y_{\mathbf{mic}}, z_{\mathbf{mic}})$**:** The 3D coordinates of the receiver.
- **Query directions (Dataset-specific):** Directional information for the source or receiver, which varies by dataset. Specifically, source direction $d_{\mathrm{src}}$ is used for the RAF dataset, and receiver direction $d_{\mathrm{mic}}$ is used for the SoundSpaces dataset.

### A.3   OUTPUT

The model's final output is a time-domain waveform of the Room Impulse Response (RIR). Internally, the model first predicts the log-magnitude Short-Time Fourier Transform (STFT) of the RIR, which is then converted to a waveform using the Griffin-Lim algorithm (Perraudin et al., 2013; Griffin & Lim, 1984). The output format is dataset-specific:

- **SoundSpaces:** Predicts binaural (stereo) RIR with receiver directional information ($d_{\mathrm{mic}}$) for discrete head orientations ($0°, 90°, 180°, 270°$).
- **RAF:** Predicts monaural RIRs with source directional information ($d_{\mathrm{src}}$).

### A.4   DATASETS

#### A.4.1   SOUNDSPACES

**Audio Data**   Following previous works (Su et al., 2022; Liang et al., 2023a;b; Brunetto et al., 2024), we use 90% of the audio data for training and 10% for testing. The Room Impulse Responses (RIRs) provided by SoundSpaces are binaural and resampled from the original 44.1 kHz to 22.05 kHz. For Short-Time Fourier Transform (STFT), we employ 512 FFT bins, with a Hann window of 512 samples and a hop length of 128 samples. All the source and microphone are lying on the same height. The source is omnidirectional.

**Visual Data**   SoundSpaces provides visual data captured via Habitat Sim (Szot et al., 2021) with RGB images of resolution $512 \times 512$ and a field of view of 90 degrees. Images for training were sampled at varying numbers depending on the size of the environment: 45 images for small rooms, 75 images for medium rooms, and 150 images for large rooms. Each training image is captured from positions randomly selected at the edges of the room, oriented towards the center with random offsets. Additionally, 50 randomly sampled test poses are used to evaluate visual performance.

#### A.4.2   RAF

**Audio Data**   Following previous works (Chen et al., 2024; Brunetto et al., 2024), we use 80% of the audio data for training and the remaining 20% for testing. Audio clips are clipped to 0.32 seconds and sampled at 48 kHz. For STFT, we employ 1024 FFT bins, a Hann window of size 512, and a hop length of 256. The source and microphone can lie on different heights. The microphone is omnidirectional, and the source is directional.

**Visual Data** Visual data in RAF are multi-view images captured using the VR-NeRF (Xu et al., 2023) camera rig Eyeful Tower. From the available 22 camera views, we select images from cameras 20 to 25 to ensure adequate overlap. Images have a resolution of $684 \times 1024$ pixels. To limit the amount of visual training data, one-third of these images are randomly subsampled, with 90% of the selected images used for training and 10% reserved for evaluation.

## B  ADDITIONAL METHOD DETAILS

### B.1  ARCHITECTURE DETAIL

#### B.1.1  NEURAL ACOUSTIC SCALING MODULE

The Acoustic Scaling Layer adjusts simulation-derived features to account for temporal scaling variations. It consists of two separate MLP branches, $\mu$ and $\nu$, each comprising two fully-connected layers with a hidden dimension of 128 and ReLU activation functions. Both branches receive the normalized time query as input. The $\mu$ branch's final linear layer is initialized to output ones, effectively serving as a scaling factor. Conversely, the $\nu$ branch's final layer is initialized to zeros, initially producing no offset.

#### B.1.2  FEATURE FUSION MLP

The Feature Fusion MLP architecture used in our model closely follows the design described in NeRAF (Brunetto et al., 2024). Specifically, the network consists of two MLP blocks. The first MLP block takes as input a concatenation of encoded positional information, directional embeddings, and encoded wave features obtained via a ResNet. This block comprises 5 fully-connected layers, each followed by Leaky ReLU activations with a slope 0.1, resulting in an intermediate 512-dimensional representation. The second MLP block processes this intermediate representation to predict the STFT bins corresponding to time queries, featuring separate output heads per audio channel. The final layer utilizes a scaled tanh activation function to appropriately map the output to the log-magnitude STFT range.

### B.2  IMPLEMENTATION DETAIL

#### B.2.1  HYPERPARAMETERS

We utilize the Adam optimizer with $\beta_1 = 0.9$, $\beta_2 = 0.999$, and $\epsilon = 10^{-15}$. The initial learning rate is set to $10^{-4}$ and exponentially decreases to $10^{-8}$. For the first 2,000 iterations, only the NeRF component is trained to allow adequate initialization. The training batch sizes are 4,096 rays for NeRF and 1,024 STFT bins for the audio model. The total training duration is 500,000 iterations, selecting the best-performing model across these iterations. Training is conducted on a single RTX A6000 GPU. These hyperparameters remain consistent across both SoundSpaces and RAF datasets.

#### B.2.2  LEARNING OBJECTIVE DETAIL

We set $\lambda_A = 10^{-3}$, $\lambda_{SC} = 10^{-1}$, and empirically set $\lambda_{SL}$ to 1 for SoundSpaces and 3 for RAF.

### B.3  FDTD WAVE SIMULATION DETAIL

#### B.3.1  SPONGE LAYER DETAIL.

We implement a sponge layer $\gamma_{\text{sponge}}$ to mitigate boundary reflections during simulation. This sponge layer has a thickness of 8 grid points, where damping coefficients gradually increase from $\gamma_{\text{min}} = 0.9$ at the outer boundary to 1.0 towards the interior, following a quadratic profile.

#### B.3.2  INITIAL AMPLITUDE SETTING.

The initial amplitude $A_{\text{init}}$ for each batch is adjusted based on the vertical distance between the source and microphone positions. This adjustment accounts for the natural attenuation of sound

energy with increasing vertical distance. Specifically, the amplitude $A_{\text{init}}$ is scaled exponentially by $\exp(-\alpha \cdot |z_{\text{src}} - z_{\text{mic}}|)$, where $\alpha$ is set to 0.01.

### B.4 FIRST-ORDER SYSTEM FORMULATION FOR FDTD

The FDTD scheme presented in Equations (1)-(5) originates from the physically-motivated first-order system of the wave equation rather than direct discretization of the second-order form. We provide the complete mathematical derivation connecting the continuous wave equation to our discrete FDTD formulation.

#### B.4.1 FROM SECOND-ORDER TO FIRST-ORDER SYSTEM

Starting from the acoustic wave equation in Equation (1), we reformulate it as a first-order system based on fundamental conservation laws:

**Conservation of momentum** (linearized Euler equation):

$$\rho_0 \frac{\partial \mathbf{v}}{\partial t} + \nabla p = 0 \quad \Rightarrow \quad \frac{\partial \mathbf{v}}{\partial t} = -\frac{1}{\rho} \nabla p \tag{20}$$

**Conservation of mass** with equation of state:

$$\frac{\partial p}{\partial t} = -\rho c^2 \nabla \cdot \mathbf{v} \tag{21}$$

where $\mathbf{v}$ is the particle velocity vector, $p$ is acoustic pressure, $\rho$ is the medium density, and $c$ is the speed of sound.

#### B.4.2 FINITE-DIFFERENCE DISCRETIZATION

We discretize this first-order system using staggered grids with spatial steps $\Delta x, \Delta y$ and time step $\Delta t$:

**Step 1 - Pressure Gradient Computation:**

$$\left.\frac{\partial p}{\partial x}\right|_{i+1/2,j} \approx \frac{p_{i+1,j} - p_{i,j}}{\Delta x}, \quad \left.\frac{\partial p}{\partial y}\right|_{i,j+1/2} \approx \frac{p_{i,j+1} - p_{i,j}}{\Delta y} \tag{22}$$

**Step 2 - Velocity Update:**

$$v_x^{n+1} = v_x^n - \frac{\Delta t}{\rho_{i,j}\Delta x}\frac{\partial p}{\partial x}, \quad v_y^{n+1} = v_y^n - \frac{\Delta t}{\rho_{i,j}\Delta y}\frac{\partial p}{\partial y} \tag{23}$$

**Step 3 - Velocity Divergence:**

$$\nabla \cdot \mathbf{v}|_{i,j} \approx \frac{v_{x,i,j} - v_{x,i-1,j}}{\Delta x} + \frac{v_{y,i,j} - v_{y,i,j-1}}{\Delta y} \tag{24}$$

**Step 4 - Pressure Update:**

$$p^{n+1} = p^n - \rho_{i,j}c_{i,j}^2\Delta t(\nabla \cdot \mathbf{v}) \tag{25}$$

The spatially-varying parameters $c_{i,j}$ and $\rho_{i,j}$ arise naturally from discretizing the continuous parameters $c(\mathbf{X})$ and $\rho(\mathbf{X})$ onto the computational grid, enabling heterogeneous media modeling. This first-order system is mathematically equivalent to the original second-order wave equation but provides a computationally stable framework for time-stepping with proper boundary condition handling.

## C ADDITIONAL EXPERIMENTAL RESULTS

### C.1 PER-SCENE RESULTS

This section provides detailed per-scene evaluation results. Table 4 shows results for individual scenes in the SoundSpaces dataset, Table 5 presents results for the RAF dataset.

| Method | scene | T60 ↓ | C50 ↓ | EDT ↓ |
|---|---|---|---|---|
| WavNAF | Apartment 1 | 2.306 | 0.446 | 0.0119 |
| | Apartment 2 | 2.400 | 0.478 | 0.0111 |
| | FRL Apartment 2 | 2.294 | 0.323 | 0.0105 |
| | FRL Apartment 4 | 2.717 | 0.262 | 0.0085 |
| | Room 2 | 0.928 | 0.254 | 0.0084 |
| | Office 4 | 1.056 | 0.239 | 0.0063 |
| | Average | 1.95 | 0.33 | 0.009 |

Table 4: Per-scene results on SoundSpaces dataset

| Method | scene | T60 ↓ | C50 ↓ | EDT ↓ |
|---|---|---|---|---|
| WavNAF | Furnished Room | 6.895 | 0.613 | 0.020 |
| | Empty Room | 7.787 | 0.598 | 0.0180 |
| | Average | 7.43 | 0.61 | 0.019 |

Table 5: Per-scene results on RAF dataset

## C.2 COMPUTATIONAL COST ANALYSIS

This section provides a detailed computational cost analysis of our method, focusing on the FDTD simulation overhead and its practical implications for training and deployment.

### C.2.1 TRAINING TIME COMPARISION

| Method | Time per Iteration | Relative Cost |
|---|---|---|
| No condition | ∼55ms | 0.6× |
| 3D grid feature | ∼94ms | 1.0× |
| Only wave feature | ∼780ms | 8.3× |
| + Acoustic scale module | ∼1010ms | 10.7× |

Table 6: Training time comparison across different model variants

Table 6 breaks down the computational overhead of different components in our pipeline. While the full method requires 10.7× more computation than the baseline, this investment yields significant quality improvements across all metrics.

The majority of overhead comes from FDTD simulation and feature extraction rather than the neural network components themselves. This suggests opportunities for optimization through parallelization or preprocessing strategies.

### C.2.2 INFERENCE TIME BREAKDOWN

- **Total inference time** : 74.74 ms
    - Preparing FDTD (acoustic parameter grid generation): 53.33 ms (71.4%)
    - Pure FDTD CUDA kernel: 3.82 ms (5.1%)
    - Other operations (feature extraction): 17.59 ms (23.5%)

Compared to NeRAF (12.36ms), our method is approximately 6× slower during inference. However, the pure FDTD computation represents only 5.1% of the total inference time, indicating that the overhead comes primarily from data preparation and feature extraction rather than from the wave simulation itself.

### C.2.3 PRACTICAL CONSIDERATIONS

While our full method requires 10.7× more training time per iteration compared to the baseline, this computational investment yields substantial improvements across all acoustic quality metrics (as

shown in Table 1 of the main paper). The computational cost is primarily front-loaded during training, with inference remaining efficient for practical deployment.

We note that our current implementation runs FDTD simulations on-the-fly during training for implementational clarity. However, these computations can be preprocessed and cached when seeking further training efficiency, potentially reducing the training overhead significantly.

## C.3 Few-shot Learning

To evaluate the data efficiency of our physics-informed approach, we conduct experiments with reduced training data. Table 7 shows performance when training with only 25%, 50%, and 75% of the available source-receiver pairs.

Our FDTD-based wave propagation priors enable significantly better generalization from limited data compared to the baseline. With only 50% of training data, WavNAF achieves comparable performance to NeRAF trained on 75% of data.

The consistent relative improvement at different data regimes indicates that our physics-informed features complement rather than replace learned representations, providing valuable guidance regardless of training set size.

| Training Fraction (%) | NeRAF | | | WavNAF | | |
|---|---|---|---|---|---|---|
| | T60 $\downarrow$ | C50 $\downarrow$ | EDT $\downarrow$ | T60 $\downarrow$ | C50 $\downarrow$ | EDT $\downarrow$ |
| 25 | 5.378 | 0.853 | 0.0226 | 4.247 | 0.726 | 0.0200 |
| 50 | 3.869 | 0.686 | 0.0180 | 3.074 | 0.553 | 0.0152 |
| 75 | 2.954 | 0.610 | 0.0158 | 2.407 | 0.501 | 0.0137 |
| 100 | **2.725** | **0.574** | **0.0149** | 2.306 | 0.446 | 0.0119 |

Table 7: Few-shot learning performance comparison at SoundSpaces apartment 1

## D Hyperparameter Ablation Study

We conduct comprehensive ablation studies to validate the robustness of our approach to various hyperparameter choices. These experiments demonstrate that our physics-informed framework provides consistent improvements regardless of specific parameter settings.

### D.1 $c_{min}/c_{max}$ Ratio

Table 8 examines the impact of varying the acoustic speed ratio in our density-to-acoustic parameter mapping (equation 12). We test ratios from 0.6 to 0.9, where lower values create stronger acoustic contrasts between air and solid boundaries. Despite these parameter variations representing different degrees of approximation from traditional acoustic modeling perspectives, all settings consistently outperform the baseline. This validates that the FDTD's physics-governed update process where wave phenomena like diffraction and interference naturally emerge from the wave equation solver provides sufficient neural guidance for acoustic field learning, regardless of exact parameter values.

| Method | $c_{min}/c_{max}$ | T60 $\downarrow$ | C50 $\downarrow$ | EDT $\downarrow$ |
|---|---|---|---|---|
| | 0.6 | 2.278 | 0.459 | 0.0123 |
| | 0.7 | **2.261** | 0.467 | 0.0120 |
| WavNAF | 0.8 | 2.377 | 0.448 | **0.0119** |
| | 0.9 (paper setting) | 2.306 | **0.446** | **0.0119** |

Table 8: Ablation study on $c_{min}/c_{max}$ ratio. Results at the SoundSpace apartment 1.

### D.2 Global Damping Coefficient

Table 9 analyzes the effect of the global damping coefficient $\gamma_{global}$, which models air absorption during wave propagation. Setting $\gamma_{global} = 1.0$ represents no damping, while $\gamma_{global} = 0.99$ introduces

gradual energy dissipation. While the presence of damping slightly improves performance, our method maintains substantial advantages over baselines even without damping, confirming that the structural information from wave physics mainly drives our performance gains.

| Method | T60↓ | C50↓ | EDT↓ |
|---|---|---|---|
| without global damping ($\gamma_{\text{global}} = 1.0$) | 2.416 | 0.456 | **0.0118** |
| with global damping ($\gamma_{\text{global}} = 0.99$) | **2.306** | **0.446** | 0.0119 |

Table 9: Ablation study on global damping coefficient. Results at the SoundSpace apartment 1.

# E    VISUALIZATIONS

## E.1    WAVE SIMULATION VISUALIZATION

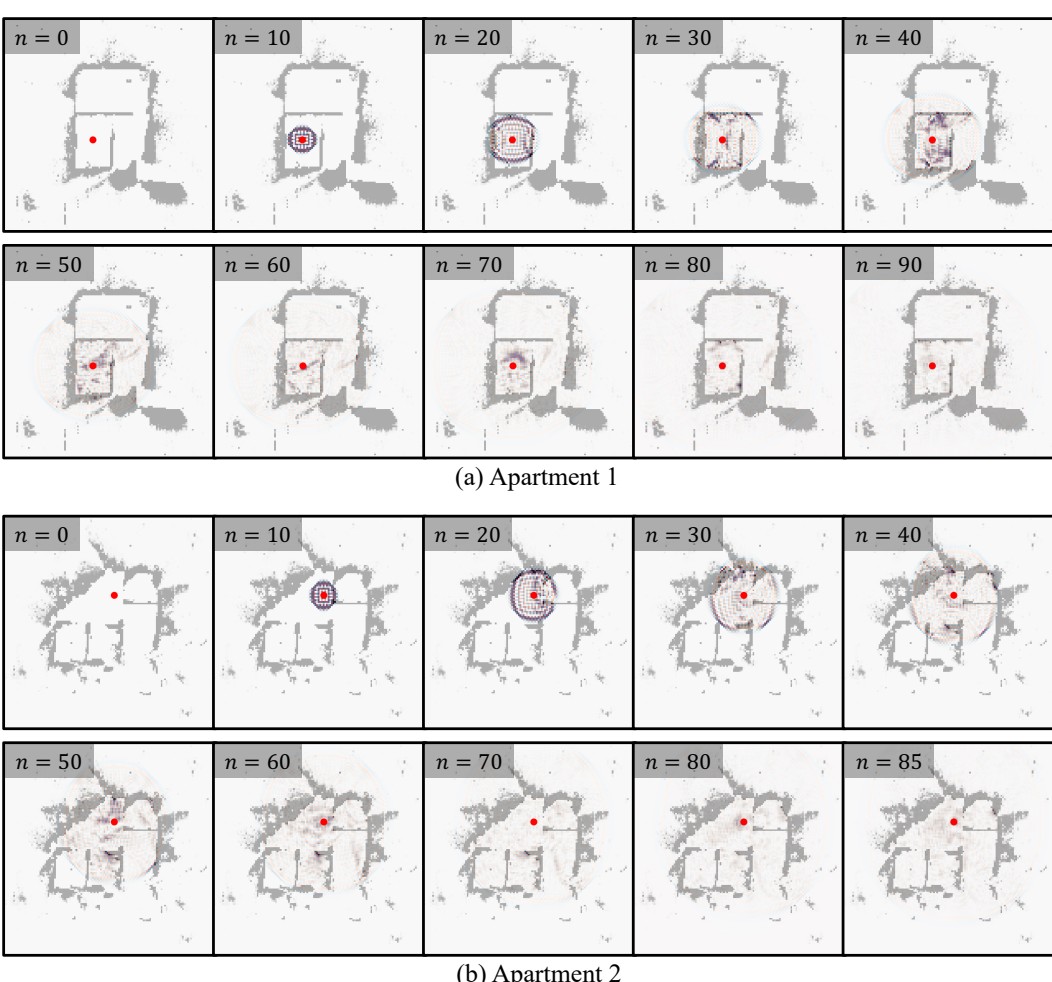

(a) Apartment 1

(b) Apartment 2

Figure 5: Wave simulation visualization of SoundSpaces dataset. Our WavNAF framework uses Neural Radiance Fields (NeRF) to extract a normalized visual density grid encoding scene geometry, from which acoustic parameters, including sound speed and acoustic density, are directly derived. This enables wave-equation-based simulations that inherently capture complex acoustic phenomena, such as diffraction, refraction, and reflection, without explicit geometric modeling.

We visualized wave simulation results for scenes with complex, multi-room layouts from the SoundSpaces dataset. A subset of frames from the full simulation sequence was selected to highlight the temporal progression of wave phenomena. These visualizations clearly demonstrate intricate wave

behaviors such as reflection, refraction, and diffraction as the simulation advances over successive time steps.

### E.2 NEURAL ACOUSTIC SCALING MODULE VISUALIZATION

We visualize the adaptive scaling behavior of our Neural Acoustic Scaling Module by plotting the scaling ratio $||\hat{f}_t||_2/||f_t||_2$, as a function of normalized time, where $f_t$ represents the original feature extracted from pressure map and $\hat{f}_t = \mu_t \odot f_t + \nu_t$ represents the scaled feature.

This visualization demonstrates that our scaling module automatically learns appropriate corrections for different temporal regions of the RIR, effectively bridging the gap between computationally efficient compressed simulations and full-scale responses. The learned scaling patterns align with known characteristics of acoustic scale models, where early reflections and late reverberation require different degrees of adjustment.

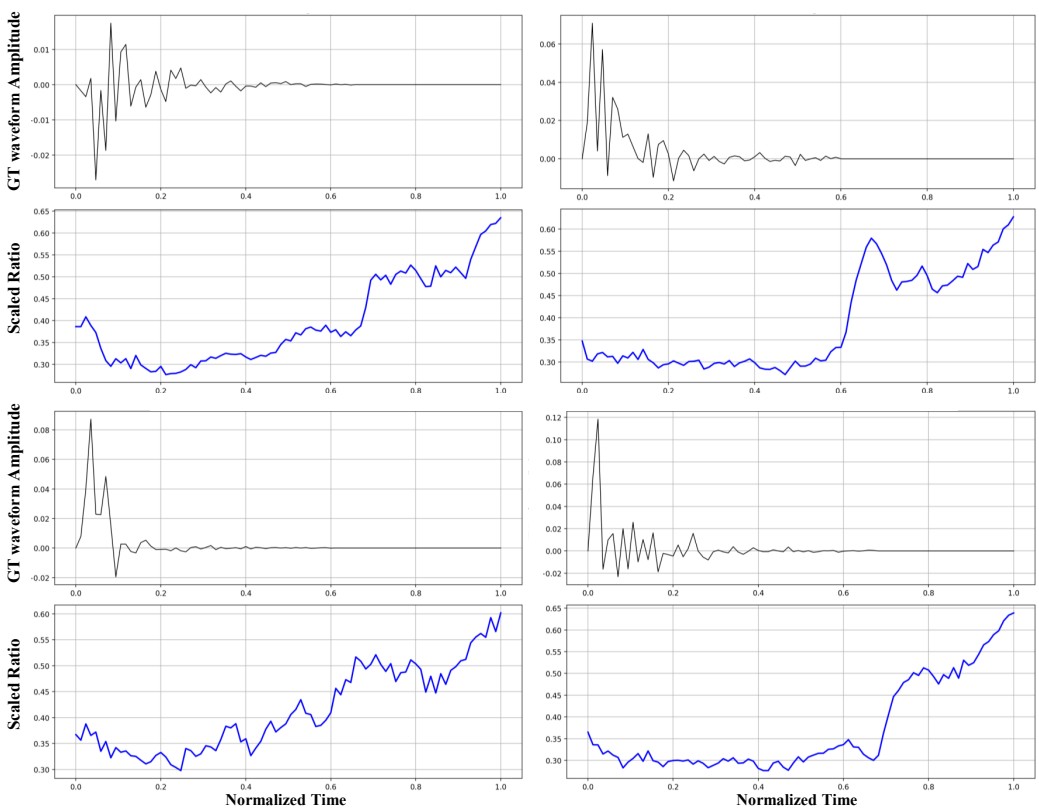

Figure 6: Visualization of the Neural Acoustic Scaling Module's adaptive behavior. The plots show ground-truth waveform amplitude (top row) and the scaling ratio $||\hat{f}_t||_2/||f_t||_2$ (bottom row) over normalized time. The scaling module learns to apply time-dependent transformations that vary between early reflections and late reverberation, adapting the features from temporally compressed simulations to match full-scale acoustic responses.

### E.3 RIR VISUALIZATION

We visualize the predicted RIRs under various source-microphone configurations in the largest scene of the SoundSpaces dataset, Apartment 1. This scene contains multiple interconnected rooms, allowing us to evaluate challenging setups where the source and microphone are positioned in different rooms. Even in a distant room separated by two doors, WavNAF accurately predicts the RIR, capturing the early-reflection spike characteristics more faithfully than competing methods. We additionally visualize the simulated pressure maps and the predicted loudness maps using the same source positions as in Figure 7. The loudness map is obtained by fixing the source position and

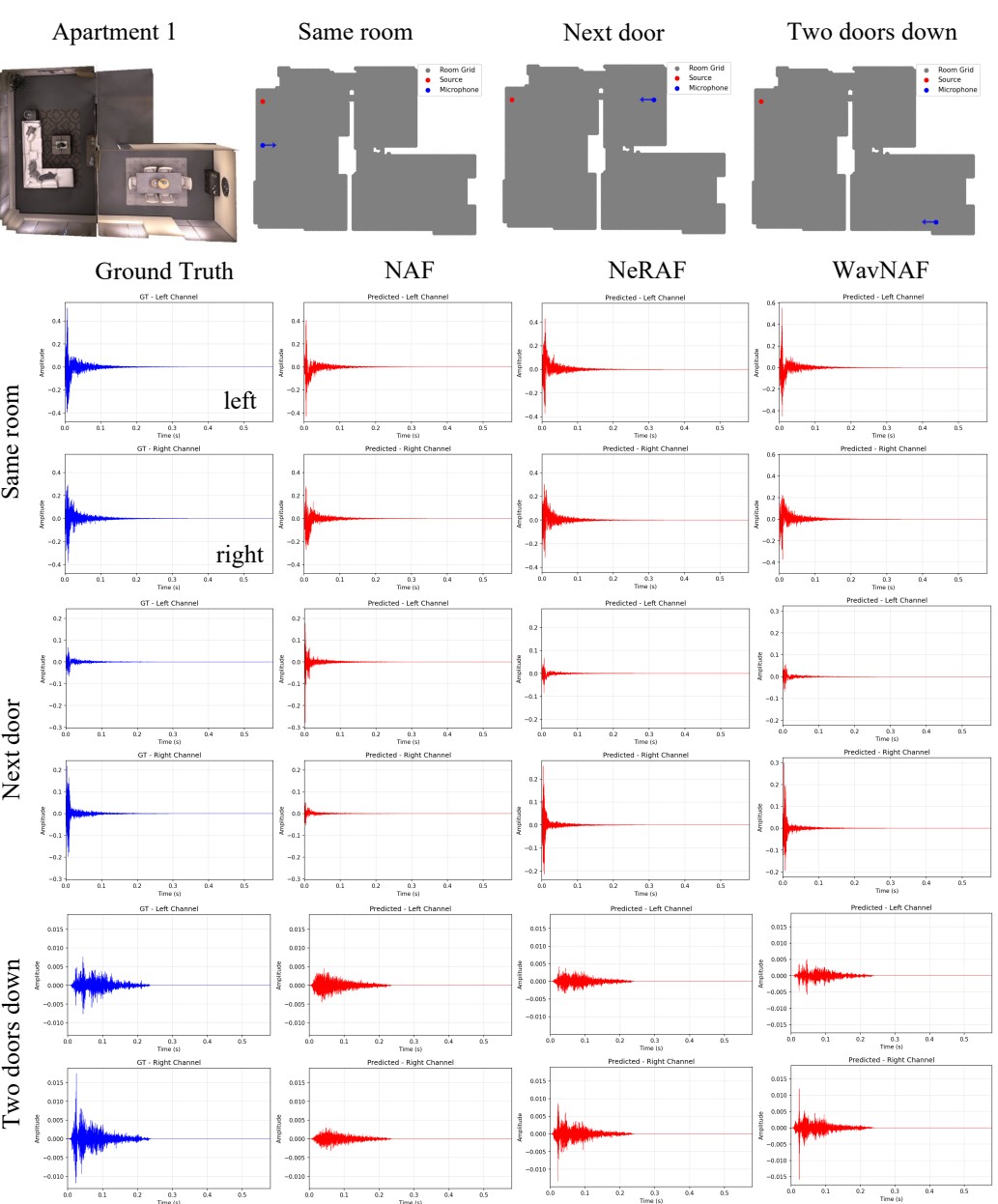

Figure 7: Predicted RIR visualization under various source-microphone configurations. Experiments are conducted in Apartment 1 of the SoundSpaces dataset. We visualize scenarios where the source and microphone are located in the same room as well as in different rooms. The farther and cross-room configurations are particularly challenging because the early part of the RIR involves multiple reflections through doors and walls. We compare WavNAF with NAF and NeRAF, and observe that WavNAF consistently outperforms both baselines across all settings. Notably, in distant room setups, WavNAF better preserves the early reflection spike characteristics, reflecting more accurate wave propagation behavior.

performing inference at all navigable points as receivers. As shonw in Figure 8 and Figure 9, the simulated pressure map and the predicted loudness map exhibit noticeable differences, primarily due to the temporal scale mismatch between the simplified simulation and the actual RIR.

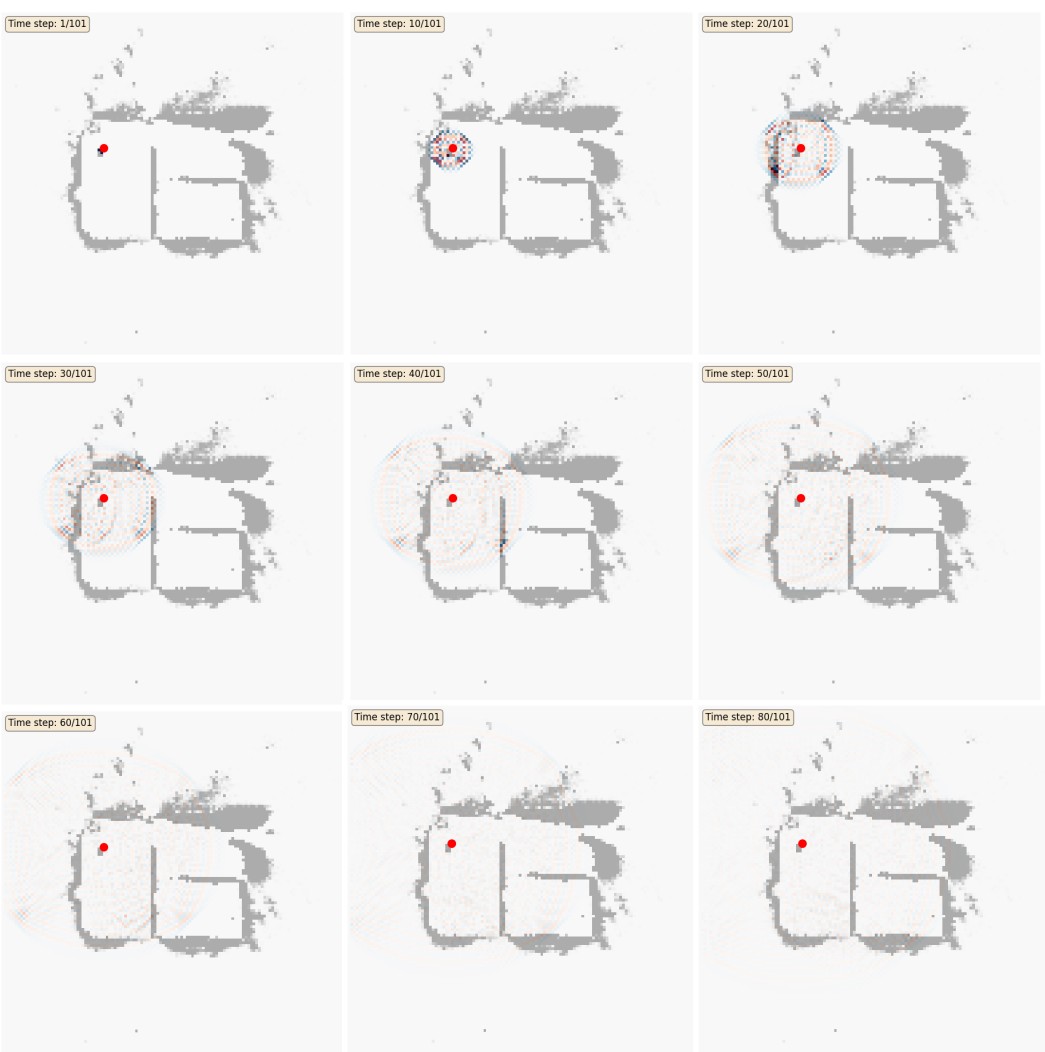

Figure 8: Visualization of simulated pressure maps. Uses same source position with Figure 7.

### E.4 Affect of source–microphone height differences

Figure 10 shows performance distribution with respect to source–microphone height differences in the Furnished Room scene of the RAF dataset.

## F The Use of Large Language Models (LLMs)

In this study, LLMs were used for text editing, grammar correction, and coding assistance for visualization.

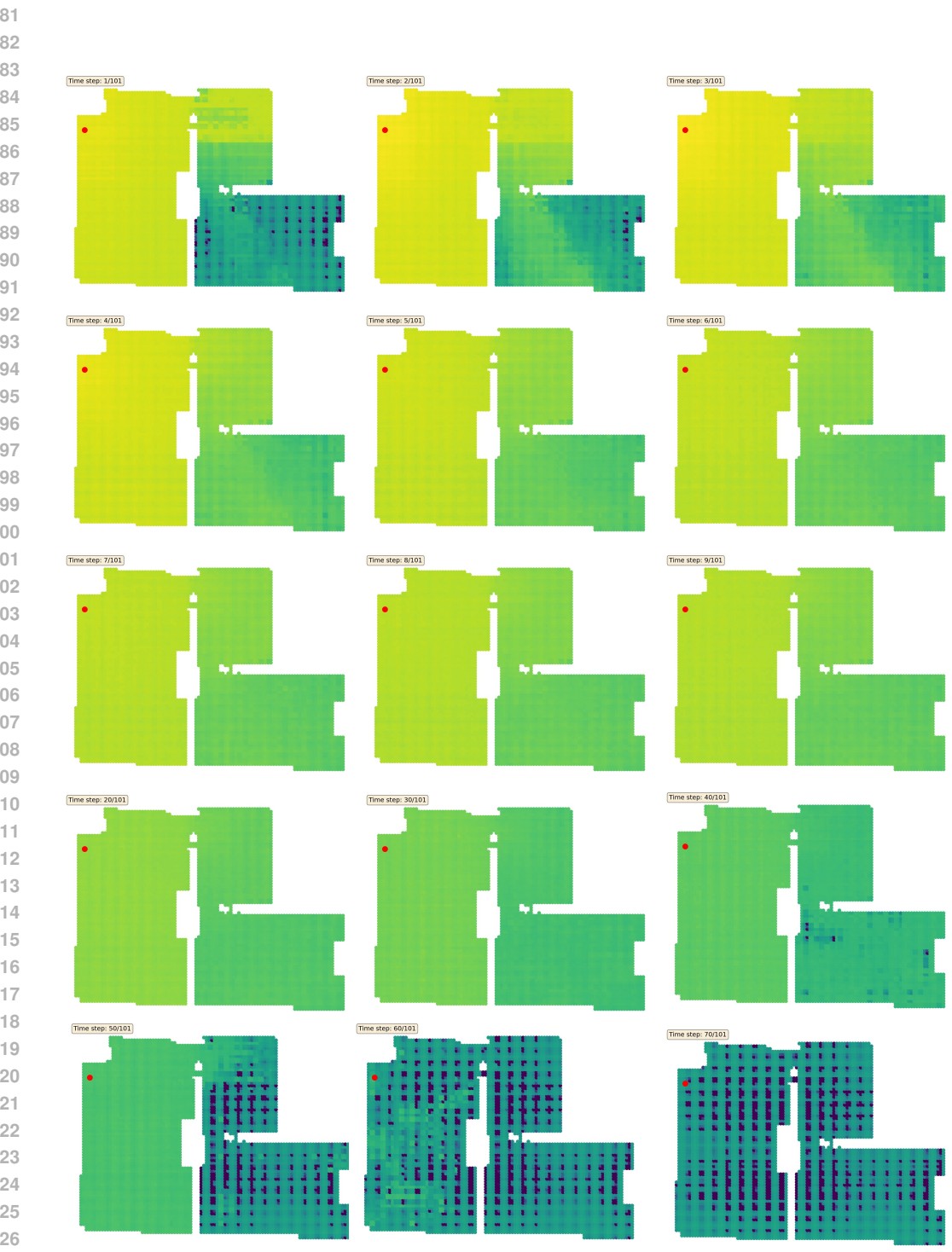

Figure 9: Visualization of loudness maps. Uses the same source position as Figure 7. The loudness map is obtained by fixing the source position and performing inference at all navigable points as receivers.

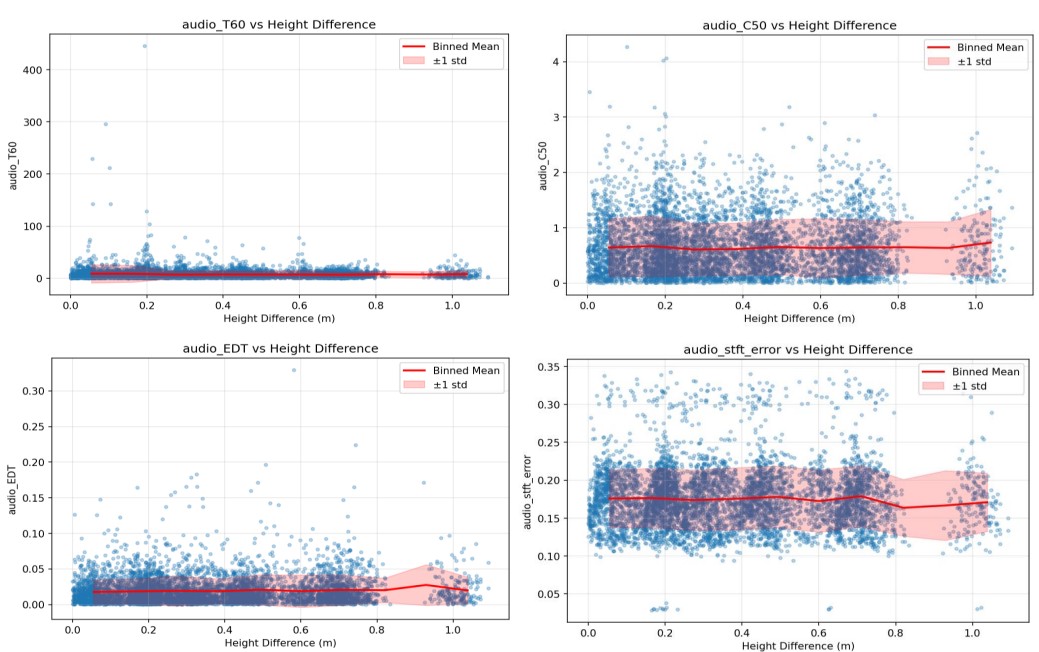

Figure 10: Performance distribution with respect to source–microphone height differences in the Furnished Room scene of the RAF dataset.

