# OpenReview forum: "WavNAF: Learning Wave Propagation Priors for Neural Acoustic Fields"
_ICLR.cc/2026/Conference — Submitted to ICLR 2026_

### Official Review · Reviewer_44Hp · 2025-10-29

**Soundness:** 2
**Presentation:** 3
**Contribution:** 3
**Rating:** 4
**Confidence:** 3

**Summary:**

The paper proposes WavNAF, which uses a NeRF-derived density field to construct 2D acoustic parameter maps and then runs a physics-based Finite-Difference Time-Domain (FDTD) simulation to produce pressure maps. A neural network then synthesizes the acoustic impulse response using features extracted from these pressure maps, along with source/microphone positions, directions, and target time. To reduce computational cost, a time-scaling module is introduced. Experiments on standard benchmarks indicate improvements over baselines.

**Strengths:**

The paper novelly uses FDTD pressure maps as inputs to explicitly encode reflection, refraction, and diffraction, injecting physical priors and reducing the burden on the network learning.

The proposed method achieved better performance than prior approaches on the evaluated datasets.

The structure of the paper is well designed, and the notations are well defined, making it easy to follow.

**Weaknesses:**

Some components are underspecified (see Questions), which complicates the full understanding of the method.

The method relies heavily on empirical choices like the mapping from optical opacity to acoustic parameters, but lacks thorough ablations to justify their ranges and robustness.

The method only considers a 2D structure when synthesizing acoustic signals, which means the full room structure is not considered in the signal synthesis.

**Questions:**

1. The paper adopts NeRF for dense reconstruction. Is any post-processing applied to the obtained opacities? Additionally, I am curious if a scene mesh is provided and then transformed to spatial grids, where the opacity is set to 1 if occupied, will it outperform the current method in theory? Also, are the areas with low optical opacity (between 0 and 1) of great importance? If not, will it achieve a better performance if you remove low-opacity areas? The visual model and the acoustic model are trained together. Am I correct in understanding that the gradient from the acoustic model does not propagate back to the NeRF densities?


2. To obtain the 2D input for simulation, the authors apply max pooling along the z-axis of the 3D density volume. However, it is unclear whether the ablation studies related to ∆z (e.g., in Table 3) are conducted on RAF, SoundSpaces, or both. Since they have different microphone/speaker setups as mentioned in the paper, it is important to know the experiment setup for robustness. Additionally, how sensitive are the results to the choice of height (a different height than the microphone height) or the grid size? Based on the visualizations, it seems that large objects such as tables, chairs, or even walls can be excluded on the selected height. Can the authors quantify how often this happens and discuss whether it affects performance across datasets? Also, please clarify the unit and value range used for ∆z in Table 3.


3. The mapping from optical opacity to acoustic parameters such as sound speed and density is empirical. This assumes that visually opaque regions correspond to acoustically reflective surfaces, regardless of material type, which is often not true. Did the authors explore other mappings, or are there any theory justifications?


4. Despite these approximations, the proposed approach achieves stronger results than prior baselines. Is this primarily because the pressure maps, even if approximate, already encode most of the wave-based propagation phenomena, thereby reducing the burden on the network learning? If so, this implies that the correctness of the pressure maps may not be strictly necessary for performance. It would help to include analysis or examples where inaccurate pressure maps lead to performance degradation, to support or challenge this hypothesis.


5. Finally, regarding Figure 6, it is unclear which comparisons directly support the claim that the proposed method bridges the gap between simulation-based and learning-based models. Could the authors provide a clearer explanation, which would be helpful?

---

> ### Author Response · Authors · 2025-11-26
> **Question 1**
>
> # (1) Post-processing of opacities
>
> No post-processing is applied. We directly use the predicted alpha 3D in Equation (10).
>
> # (2) Would a ground-truth mesh-based grid outperform the current method?
> To verify this, we constructed a ground-truth spatial grid from the scene meshes provided in the RAF dataset. Using this grid, we obtained:
>
> T60: 7.18 (paper: 7.43)
>
> C50: 0.59 (paper: 0.61)
>
> EDT: 0.018 (paper: 0.019)
>
> STFT error: 0.16 (paper: 0.16)
>
> # (3) Importance of low-opacity regions
> We further binarized the NeRF density by setting opacity < 0.5 to 0 and ≥ 0.5 to 1. The results are:
>
> T60: 7.47 (paper: 7.43)
>
> C50: 0.61 (paper: 0.61)
>
> EDT: 0.019 (paper: 0.019)
>
> STFT error: 0.16 (paper: 0.16)
>
> This binary grid yields a slightly improved T60.
>
> # (4) Do gradients propagate from the acoustic model to NeRF?
> No. The acoustic model is trained independently from the NeRF densities, and gradients do not flow back to the visual model.

---

> ### Author Response · Authors · 2025-11-26
> **Question 2**
>
> # (1) Dataset used in Table 3
> As stated in the caption, the ablations in Table 3 are conducted on the SoundSpaces dataset.
>
> # (2) Sensitivity to source–microphone height differences
> We added an additional visualization in Figure 10, showing the performance distribution with respect to source–microphone height differences. Across different height offsets, the mean values of T60, C50, and EDT remain stable, indicating that the performance is not strongly affected by moderate height mismatches.
>
> # (3) How often large objects are excluded and whether this impacts performance
> We agree with the reviewer that in some configurations, large objects such as tables or tall chairs may be partially excluded from the selected height slice. However, this depends on the specific source–microphone placement and furniture configuration in each scene. As these layout configurations are defined in the RAF dataset itself, we refer the reviewer to the original RAF dataset description for detailed statistics. Empirically, as shown in Figure 10, performance degradation is minimal even in cases with noticeable vertical offsets, suggesting that the model is robust to such occurrences.
>
> # (4) Unit and value range of Δz in Table 3
> Δz denotes the difference in the number of discrete cells along the z-dimension of the 3D grid. Therefore, it is a unitless integer value.

---

> ### Author Response · Authors · 2025-11-26
> **Question 3**
>
> We thank the reviewer for raising this important point. Our current mapping from optical opacity to acoustic parameters is indeed empirical, based on the assumption that visually opaque regions generally correlate with acoustically reflective surfaces. We acknowledge that this does not account for detailed material properties, which may vary across scenes.
>
> However, although this mapping is simplified, its effect on acoustic performance is relatively minor in our framework. As shown in Appendix D, across a variety of simulation-related hyperparameter settings, including those that implicitly modify the effective reflectivity of surfaces, the performance variation remains small. This empirical observation suggests that our method is robust to the exact mapping between opacity and acoustic material parameters.
>
> Exploring richer mappings that incorporate material semantics or learned acoustic reflectivity is an exciting future direction, and we believe our framework can naturally support such extensions.

---

> ### Author Response · Authors · 2025-11-26
> **Question 4**
>
> We appreciate the reviewer’s thoughtful hypothesis. Our findings indeed align with the idea that better pressure maps can lead to improved performance. As shown in our ground-truth (GT) grid experiment reported in Q1, using the GT occupancy grid yields higher performance than using the learned grid. This confirms that more accurate pressure maps provide more reliable wave propagation cues that the network can effectively leverage.
>
> At the same time, the results in Appendix D demonstrate that our framework is robust to inaccuracies in the simulated pressure maps. Perturbing the simulation parameters produces slightly degraded pressure maps, which results in some performance drop. However, the degradation remains relatively small across all acoustic metrics. This suggests that while high-quality pressure maps are beneficial, the correctness of the maps does not need to be perfect for the model to perform well.

---

> ### Author Response · Authors · 2025-11-26
> **Question 5**
>
> As described in Sections 3.2 (Acoustic Scale Model) and 4.3 (Neural Acoustic Scaling Module for Temporal Alignment), there exists a significant temporal mismatch between FDTD simulation time and the actual RIR duration. For the “Apartment 1” scene, a CFL-constrained FDTD step size is 0.0002 seconds, and 101 steps simulate only about 0.02 seconds, which is roughly 3% of the actual RIR length (approximately 0.6 seconds). This means that the raw simulation covers only a very short portion of the true temporal propagation window and is therefore insufficient to model late reverberation directly.
>
> To address this gap, the Neural Acoustic Scaling Module adaptively adjusts the feature scaling over time, compensating for the limited temporal depth of simulation. As highlighted in Sections 3.2 and 4.3, early reflections and late reverberation require different adjustment strengths, and we hypothesize that stronger scaling is especially needed for the late reverberation region, where FDTD simulation provides no physical evidence.
>
> Figure 6 directly supports this hypothesis: the learned scaling factors progressively increase toward the later part of the time axis, indicating that the model relies more heavily on learned corrections as it transitions from simulation-dominated early reflections toward neural modeling of late reverberation. This behavior visually and quantitatively demonstrates how our method bridges the gap between simulation-based and learning-based approaches.

---

### Official Review · Reviewer_3MYx · 2025-10-30

**Soundness:** 3
**Presentation:** 3
**Contribution:** 3
**Rating:** 8
**Confidence:** 4

**Summary:**

The paper presents WavNAF, designed to improve room acoustics modeling by integrating wave propagation physics directly into the acoustic synthesis process. It overcomes the limitations of geometric methods, which cannot model intricate wave phenomena like diffraction, refraction, and complex reflections, by utilizing the Finite-Difference Time-Domain (FDTD) method to numerically solve the wave equation.
The key contributions include generating physically-informed wave propagation priors by deriving acoustic parameters (sound speed and density) from visual scene geometry encoded by NeRF. Crucially, the framework introduces a Neural Acoustic Scaling Module that efficiently addresses the high computational cost of full FDTD simulations by learning adaptive, time-dependent transformations to accurately estimate full-scale RIRs from compressed simulations. This physics-based approach leads to substantial improvements in acoustic quality across standard metrics compared to existing state-of-the-art methods.

**Strengths:**

- WavNAF integrates physically-informed wave propagation priors derived directly from FDTD-simulated pressure maps, providing a strong inductive bias for neural acoustic field learning, even with simplified material parameters.
- The Neural Acoustic Scaling Module addresses the high computational cost and stability constraints of FDTD by learning adaptive, time-dependent transformations to estimate accurate full-scale RIRs from temporally compressed simulations.
- The physics-informed wave propagation priors enable better generalization from limited data
- Using NeRF to extract visual scene geometry, converting volumetric density fields into 2D acoustic parameter grids (sound speed and density) allows physics-informed simulations without explicit material annotations
- The paper is well-written, and I really like the detailed background section.

**Weaknesses:**

- WavNAF shows sensitivity to the alignment between the visual and acoustic coordinate systems. The model currently requires empirical tuning to ensure the necessary precise coordinate alignment between the geometry derived from NeRF and the simulation space
- The integration of on-the-fly FDTD simulations significantly increases the computational complexity compared to purely neural or geometric baselines
- To model energy dissipation and numerical stability, the framework uses a combination of an empirically defined sponge layer near the boundaries and a simple global damping factor applied uniformly across the domain to model natural air attenuation. While effective, this is a simplified model of complex, frequency-dependent air absorption effects.

**Questions:**

- Given that the FDTD CUDA kernel accounts for only 5.1% of the total inference time, and data preparation (acoustic parameter grid generation) accounts for 71.4%, what specific optimizations are being considered for the data preparation pipeline to make the transition to full 3D FDTD simulations feasible? Besides, if computational resources were unlimited, what is the potential performance gain anticipated by moving from 2D to 3D simulations? In particular, how would the inclusion of vertical propagation specifically impact the accuracy of metrics like EDT and C50?
- Have the authors experimented with different initialization strategies for the MLPs that are specifically designed to emphasize correction for late reverberation, such as initializing $\nu$ with a slight non-zero bias, and how did this affect convergence stability?
- The few-shot learning study shows that WavNAF achieves comparable performance to a baseline (NeRAF) trained on 75% of data when WavNAF is trained on only 50% of data. Does this data efficiency hold true when testing on the RAF real-world dataset, where the noise and complexity may be greater than in the synthetic SoundSpaces data?

---

> ### Author Response · Authors · 2025-11-26
> **Question 1**
>
> Currently, several key operations still run on Python: (i) slicing the target 2D plane from the 3D grid for each element in the batch, (ii) applying the acoustic parameters on this sliced plane, and (iii) mapping continuous source and receiver positions represented in floating-point coordinates into discrete grid indices. Only the FDTD step updating is implemented in CUDA. We are therefore exploring fused CUDA kernels for these operations to eliminate Python-side loops and minimize host-device synchronization. We expect this will substantially reduce the dominant data preparation cost and improve the efficiency of our current 2D FDTD simulation, while also making the pipeline more scalable toward future 3D extensions.
>
> On the other hand, extending the solver from 2D to 3D introduces a fundamentally different scale of computation. Ignoring stencil pattern complexity, moving from an N x N grid to an N x N x N grid already introduces an N-fold increase in the number of cells. In addition, the CFL constraint becomes stricter in 3D, increasing the number of required time steps by roughly 1.22 times. The number of acoustic field variables also increases from (p, v_x, v_y) to (p, v_x, v_y, v_z), which adds another factor of around 1.33. As a result, the total computational cost increases by approximately N x 1.22 x 1.33 ≈ 1.63N. For a typical resolution of N = 128, this represents more than a 200 times increase over the 2D case, and this estimate is conservative since it does not include the additional overhead of 3D stencils and memory bandwidth limitations.
>
> If computational resources were unlimited and full 3D FDTD simulation were feasible, the benefits would extend well beyond simply incorporating vertical propagation. Three-dimensional propagation would allow accurate modeling of how acoustic waves interact with furniture and other elements with substantial vertical structure, such as tables, shelves, and sofas, which are only roughly projected onto the 2D plane in our current setup. Many critical early reflections, including occlusion and scattering effects caused by such objects, inherently involve vertical energy transport. Since EDT and C50 are highly sensitive to both the strength and timing of these object-dependent early reflections, capturing their true 3D propagation paths would yield more reliable estimates of early decay behavior and the balance between early and late arriving energy.

---

> ### Author Response · Authors · 2025-11-26
> **Question 2**
>
> We really thank the reviewer for pointing out the issue regarding the MLP initialization. Following the feedback, we replaced our original MLP initialization with Kaiming He initialization and re-evaluated WavNAF on the RAF dataset. With this change, we obtained the following performance:
>
> * T60: 7.36 (NeRAF: 7.47, previous version WavNAF: 7.43)
> * C50: 0.59 (NeRAF: 0.61, previous version WavNAF: 0.61)
> * EDT: 0.018 (NeRAF: 0.020, previous version WavNAF: 0.019)
> * STFT error: 0.16 (NeRAF: 0.17, previous version WavNAF: 0.16)
>
> We have removed all mentions of the previous initialization scheme in the revised paper to avoid confusion.
> Although we could not rerun all experiments in the paper due to the limited rebuttal timeline, we confirm that this initialization leads to improved performance. We will update all experiments with the improved initialization for the final camera-ready version.

---

> ### Author Response · Authors · 2025-11-26
> **Question 3**
>
> As shown in the table below, WavNAF trained with only **50%** of the data achieves an EDT of **0.0206**,
> which is slightly better than NeRAF trained with **75%** of the data (EDT = 0.0210).
> Since EDT reflects early-reflection quality, this indicates that WavNAF can
> recover early reflections reliably even with less training data.
>
> | Train Ratio | NeRAF T60 | NeRAF EDT | NeRAF C50 | NeRAF STFT | WavNAF T60 | WavNAF EDT | WavNAF C50 | WavNAF STFT |
> |------------:|----------:|----------:|----------:|-----------:|-----------:|-----------:|-----------:|------------:|
> | 25%         | 9.8157    | 0.0266    | 0.8696    | 0.1694     | 9.5127     | 0.0236     | 0.7511     | 0.1727      |
> | 50%         | 8.5270    | 0.0241    | 0.7226    | 0.1737     | 8.3505     | 0.0206     | 0.6607     | 0.1726      |
> | 75%         | 8.0687    | 0.0210    | 0.6540    | 0.1702     | 7.7141     | 0.0192     | 0.6201     | 0.1667      |

---

### Official Review · Reviewer_k2CB · 2025-10-30

**Soundness:** 2
**Presentation:** 3
**Contribution:** 2
**Rating:** 4
**Confidence:** 4

**Summary:**

This paper presents WavNAF, a hybrid physics-guided/neural framework for room-acoustics synthesis. First, a NeRF is trained from multi-view images and its volumetric density is collapsed into a two-dimensional floor-plane grid that is heuristically mapped to local sound-speed and air-density values. A 2-D FDTD solver then generates a short sequence of pressure maps conditioned on a point source, after which two small MLPs learn an affine scaling transform so that these time-compressed simulations can still predict full-scale RIR short-time Fourier transform bins. Finally, positional encodings of the source/listener and directional queries are fused with the scaled pressure features in a neural field that regresses log-magnitude STFT frames. Experiments on SoundSpaces and RAF show that WavNAF attains slightly better T60, C50, EDT and log-STFT error than baselines such as NAF, INRAS, NeRAF and AV-GS while requiring substantially shorter simulations than full-length FDTD.

**Strengths:**

1. Physics-aware prior. Incorporating FDTD pressure maps provides an interpretable link between geometry and acoustics and yields consistent gains in ablations.
2. The neural scaling module reduces simulation length 18× with < 2 % drop in metrics, which is important for interactive RIR prediction.

**Weaknesses:**

1. 2-D pressure field. All wave simulations are confined to a floor-plane slice, ignoring vertical propagation, ceiling reflections and height-dependent diffraction.
2. Modest improvements on RAF. Relative gains over NeRAF are marginal. (Compared with NeRAF T60 7.47 -> 7.43, C50 0.61 -> 0.61)
3. Missing qualitative visualisations. No figures compare simulated versus learned pressure fields, making it hard to judge fidelity.

**Questions:**

1. Can you provide more informative visualization on the learned acoustic field? Like your simulated pressure field and the final result. I noticed you included some visualization in the appendix. However, that figure is just hard to understand.
2. It seems that the pressure map simulation already estimate the impulse response. Why didn't you directly user that output? If the acoustic feature is accurately estiamted. Then I would imagine you do not need a subsequent NAF model to estimate the STFT.
3. What is the time cost of the simulation if you want to simulate a single sample, for example, fix a tx and rx locations.
4. If you already use FDTD method to simulate the pressure field and you know the global geometries. I would assume that these simulated data would already good enough to represent the whole acoustic field. So if you only use 10% dataset to train the dataset for example on RAF dataset, would you have much better performance?
5. What is the reason you predict the STFT rather than the impulse resposne in the time domain? It seems that you can already simulate the pressure field in the small time axis, is it because the computation budget?

Besides, I would encourage you to cite more recent works on the related topics:
- Jin, Derong, and Ruohan Gao. "Differentiable Room Acoustic Rendering with Multi-View Vision Priors." arXiv preprint arXiv:2504.21847 (2025).
- Lan, Zitong, Yiduo Hao, and Mingmin Zhao. "Resounding Acoustic Fields with Reciprocity." arXiv preprint arXiv:2510.20602 (2025).
- Liang, Susan, Chao Huang, Yunlong Tang, Zeliang Zhang, and Chenliang Xu. "p-AVAS: Can Physics-Integrated Audio-Visual Modeling Boost Neural Acoustic Synthesis?." In Proceedings of the IEEE/CVF International Conference on Computer Vision, pp. 13942-13951. 2025.

---

> ### Author Response · Authors · 2025-11-26
> **Comments on Weaknesses**
>
> Following Reviewer 3MYx’s feedback, we replaced our original MLP initialization with Kaiming He initialization and evaluated WavNAF on the RAF dataset. With this change, we obtained the following performance:
>
> * T60: 7.36 (NeRAF: 7.47)
> * C50: 0.59 (NeRAF: 0.61)
> * EDT: 0.018 (NeRAF: 0.020)
> * STFT error: 0.16 (NeRAF: 0.17)
>
> Due to the limited rebuttal time, we could not rerun all experiments in the paper with the updated initialization. However, we confirm that this refined initialization leads to performance improvement, and we will update all experiments with the improved initialization in the final camera-ready version.

---

> ### Author Response · Authors · 2025-11-26
> **Question 1**
>
> We added more visualizations for various scenarios. Please see Appendix E.3 and the supplementary material zip file. It includes a comparison between the simulated pressure field and the predicted loudness map.

---

> ### Author Response · Authors · 2025-11-26
> **Question 2**
>
> In principle, we agree with the reviewer that, given an accurate FDTD simulation, one can obtain an impulse response at a receiver by accumulating the simulated pressure at that grid point over time. However, in our framework the FDTD stage is deliberately simplified and scaled down, and therefore its output is not intended to serve as a ground-truth RIR.
>
> First, our FDTD is run on a (128 × 128) 2D floor-plane grid in the normalized NeRF space, which is much coarser than the true 3D room and ignores vertical propagation and ceiling reflections. Second, to satisfy the CFL condition while keeping the simulation practical, we only simulate a short, temporally compressed window instead of the full RIR time span. Third, the acoustic parameters (ρ_map) and (c_map) are heuristically derived from NeRF density without explicit material labels or frequency-dependent absorption. Under these conditions, simply accumulating the pressure trace at a receiver cell does not yield a physically accurate impulse response for the real environment.
>
> Despite these limitations, the FDTD still provides a useful **wave propagation prior** within our framework. The state update rules (Eqs. (2)–(5)) implement the wave equation and thus encode essential propagation behavior such as wavefront curvature, diffraction around obstacles, and multi-order reflections. Even if the absolute values of ρ and c are physically inaccurate, their spatial trends (e.g., higher density in visually occupied regions) are preserved, and each time step strictly follows the underlying physical law. This makes the resulting pressure maps a valid and effective prior for our neural acoustic field, rather than a substitute for the ground-truth impulse response itself.

---

> ### Author Response · Authors · 2025-11-26
> **Question 3**
>
> The requested information is provided in Appendix C.2.2 of the manuscript.

---

> ### Author Response · Authors · 2025-11-26
> **Question 4**
>
> Another reviewer raised a similar question regarding the effect of limited training data.
> Rather than presenting only the 10% setting, we provide results across multiple training-data ratios
> to show the overall performance trend. All results are averaged over the two RAF scenes:
> Furnished Room and Empty Room.
>
> As shown in the table below, WavNAF trained with only **50%** of the data achieves an EDT of **0.0206**,
> which is slightly better than NeRAF trained with **75%** of the data (EDT = 0.0210).
> Since EDT reflects early-reflection quality, this indicates that WavNAF can
> recover early reflections reliably even with less training data.
>
> | Train Ratio | NeRAF T60 | NeRAF EDT | NeRAF C50 | NeRAF STFT | WavNAF T60 | WavNAF EDT | WavNAF C50 | WavNAF STFT |
> |------------:|----------:|----------:|----------:|-----------:|-----------:|-----------:|-----------:|------------:|
> | 25%         | 9.8157    | 0.0266    | 0.8696    | 0.1694     | 9.5127     | 0.0236     | 0.7511     | 0.1727      |
> | 50%         | 8.5270    | 0.0241    | 0.7226    | 0.1737     | 8.3505     | 0.0206     | 0.6607     | 0.1726      |
> | 75%         | 8.0687    | 0.0210    | 0.6540    | 0.1702     | 7.7141     | 0.0192     | 0.6201     | 0.1667      |

---

> ### Author Response · Authors · 2025-11-26
> **Question 5**
>
> We believe Q5 stems from a similar misunderstanding as Q2. It is correct that our FDTD solver
> can simulate pressure fields along a small time axis (which corresponds to the minimal
> simulation time length permitted by the CFL stability condition). It is also true that some
> neural acoustic models directly estimate the waveform impulse response in the time domain;
> for example, NACF predicts time-domain impulse responses.
>
> However, many recent neural acoustic field methods, including AV-NeRF, NeRAF, NAF,
> and INRAS, predict the STFT representation rather than the raw waveform. This formulation
> has already been widely adopted in prior work, and its effectiveness has been consistently
> validated in the neural acoustic field literature.

---

> ### Author Response · Authors · 2025-11-26
> **Cite more recent works**
>
> We agree with the reviewer's comment. However, due to the page limit, we will update related works in the final camera-ready version.

---

### Official Review · Reviewer_QtCm · 2025-11-02

**Soundness:** 2
**Presentation:** 3
**Contribution:** 2
**Rating:** 4
**Confidence:** 4

**Summary:**

This paper introduces a novel framework to estimate room impulse responses (RIRs) from scene geometry. It first simulates low-resolution wave propagation using the Finite-Difference Time-Domain (FDTD), aiming to capture audio propagation impacting factors such as  reflections and diffraction, then trains a Neural Acoustic Field conditioned on these physics-based priors. A Neural Acoustic Scaling Module refines frequency-dependent detail. Experiments on synthetic and real-world (Real Acoustic Fields) datasets show improved realism and generalization over prior neural methods. WAVNAF achieves physically faithful, data-efficient acoustic field reconstruction for both simulated and measured indoor environments.

**Strengths:**

1. The paper is well-presented, easy to follow, the motivation is clear as well.
2. The paper tries to solve an important problem that models spatial audio propagation process with neural network.
3. The results presented in the paper show advantage over other baselines.

**Weaknesses:**

The main weakness, based on my understanding, is the lack of clear verification to show the necessity of using vision to assist spatial audio propagation process. As the way audio propagates in an enclosed 3D space, its hebaviour is impacted by lots of factors, construction material, indoor surface aborption coefficient, reflection parameter etc.. The vision alone can't fully capture all of these influencing factors, but the authors claims in the paper that they can successfully model factors like diffraction, diffusion and reflection. I look forward more explaination and clarification on vision exploitation validation during the rebuttal period.

**Questions:**

1. L203 describing local sound speeds, what is the local sound speed? Isn't the sound speed a constant in the whole room scene?
2. One missing comparing baseline: Deep Neural Room Acoustics Primitve, ICML24; Deep Impulse Responses: Estimating and Parameterizing Filters with Deep Networks, ICASSP, 2022.
3. Lack of visualization on how the predicted RIR look like, the performance regarding room size, receiver-source distance and the impact of indoor furnitures.
4. The paper just predicts STFT map for the RIR, ignoring the phase information. As RIR data is sensitive to position change (in other words, the phase is important), the paper should at least experimentally verify why ignoring phase information is a good option.

---

> ### Author Response · Authors · 2025-11-26
> **Comments on Weaknesses**
>
> We first thank the reviewer for carefully reading our paper and for their constructive comments on the limitations of our approach. We agree that having access to a full and accurate mesh of the room geometry would enable more precise FDTD simulation and would also make it possible to use ray-tracing based methods. However, in most practical scenarios it is very difficult to obtain such detailed mesh information for a target room. In contrast, our framework is designed to operate with noisy geometry that is learned from only a small set of casually captured RGB images of the room, and still leverage this imperfect geometry for RIR estimation. The noisiness of the learned geometry can be clearly seen in the visualizations in Appendix E.
>
> Since we run FDTD on noisy geometry with simplified material assumptions, the simulated wave field is of course not identical to the real acoustics. Nevertheless, it serves as an effective wave propagation prior because each update step in the solver strictly follows the underlying physical laws. In Appendix D, we further report ablation studies on simulation-related hyperparameters (e.g., material and decay perturbations), and observe that WavNAF consistently outperforms the baselines even under such perturbations, indicating that our framework can robustly benefit from imperfect simulations. As shown in Eq. (2)–(5) in the main paper and Appendix B.4, the update scheme is directly derived from the wave equation, which encodes the intrinsic behavior of acoustic waves. As a result, phenomena such as diffraction, reflection, and refraction emerge naturally from the simulation without requiring any explicit hand-crafted modeling, even when the geometry and materials are not perfectly accurate. Importantly, faithfully capturing these physical phenomena and exactly reproducing real-world acoustics are different goals; our method explicitly targets the former. To the best of our knowledge, among neural approaches in the literature, including concurrent work, WavNAF is the only framework that explicitly accounts for diffraction and rich multiple reflections through a physics-based wave solver.

---

> ### Author Response · Authors · 2025-11-26
> **Question 1**
>
> "Local sound speeds" refers to spatially varying sound speeds, which are a well-established physical phenomenon, not an assumption.
>
> Sound speed varies with the medium. It is a fundamental principle of acoustics. Since indoor spaces consist not only of air but also walls, furniture, and various other materials, it is natural to model different sound speeds at different locations.
>
> In our method, we utilize NeRF's density information to assign sound speeds to each grid cell (Equation 12). Assigning spatially varying acoustic parameters to grid cells is a standard approach in FDTD simulations for modeling heterogeneous media (Hamilton, 2016). These spatially varying sound speeds create impedance mismatches that naturally capture wave phenomena such as reflection and diffraction.

---

> ### Author Response · Authors · 2025-11-26
> **Question 2**
>
> Thank you for pointing out *Deep Neural Room Acoustics Primitive* (DeepNeRAP) and *Deep Impulse Responses* (IR-MLP). Both are important works on neural IR modeling, but they operate under a different problem setting from WavNAF.
>
> As described in Appendix A.1 “Task definition’’ and A.2 “Required inputs’’ of our paper, WavNAF learns a scene-level RIR field conditioned on visual 3D geometry: the model takes (i) multi-view RGB images with known camera poses (for NeRF-based 3D reconstruction) and (ii) query source/receiver positions, and outputs the corresponding RIR. We do not assume access to excitation–response audio from the target environment at test time in the SoundSpaces/RAF benchmarks.
>
> IR-MLP is formulated as estimating filters in an LTI system from paired input–output audio. In the abstract (p. 1), the authors describe estimating characteristics of “an observed signal given the source signal,” and Sec. 2 “MODEL FORMULATION’’ (p. 2) defines an LTI system where a known source signal \(s(t)\) and the observed (target) signal \(y(t)\) are given and the impulse response \(h\) is estimated. Sec. 4 “EVALUATION,’’ Dataset paragraph (p. 2), further states that they synthetically generate the observed (target) signals by convolving a logarithmic sine sweep (the source signal \(s(t)\)) with each measured IR. Thus IR-MLP explicitly assumes excitation–response audio pairs in the environment whose IR is being estimated and does not use visual/geometric inputs.
>
> DeepNeRAP is built around active acoustic probing in each target room. The pipeline and Fig. 1 (p. 2) describe two agents that “actively probe the room scene acoustically by emitting and receiving sounds,’’ and the model is optimized by minimizing the discrepancy between the receiver’s recorded sound and the neural-RIR-effected sound. In Sec. 3 (around Eq. (2), p. 3), the authors state “In our setting, the source agent emits a sine sweep … so as to cover the whole frequency range,’’ and on p. 4 they describe the self-supervision signal as “the difference between the emitted sound and the received sound,’’ enforcing the predicted neural-RIR-effected sound to match the receiver-agent recorded sound. In Sec. 4.2 “Comparison Methods’’ (p. 7), they further note: “Currently there are no existing methods sharing exactly the same problem setting with our framework.’’
>
> Both IR-MLP and DeepNeRAP therefore require excitation–response audio recorded in the target environment (paired source/observed signals or actively probed emitted–received pairs). In contrast, WavNAF is evaluated under a protocol where test-time inputs are visual 3D scene information plus query source/receiver positions, without observed excitation–response audio. We therefore view IR-MLP and DeepNeRAP as complementary, audio-driven approaches, rather than direct baselines for our visual-geometry–conditioned setting.

---

> ### Author Response · Authors · 2025-11-26
> **Question 3**
>
> We added more visualizations for various scenarios. Please see Appendix E.3 and the supplementary material zip file.
>
> To clarify the dependence on room size, in the SoundSpaces benchmark Apartment 1 and Apartment 2 correspond to the largest scenes, FRL Apartment 2 and FRL Apartment 4 are medium-sized rooms, and Room 2 and Office 4 are small rooms. For detailed room geometry and layouts, we refer the reviewer to the Replica dataset paper. Consistent with intuition, we observe that smaller rooms (Room 2, Office 4) are generally easier to model, which is reflected by slightly better quantitative metrics compared to larger apartments. Please see results in Appendix C.1.
>
> For the impact of indoor furniture, please refer to the RAF benchmark results. The RAF dataset provides paired Furnished Room and Empty Room scenes that share identical room and differ only in the presence of furniture. As reported in the per-scene results in Appendix C.1, WavNAF achieves higher T60 accuracy in the furnished scenes, whereas C50 and EDT are better in the empty scenes. Since T60 mainly reflects the modeling of late reverberation and EDT focuses on early reflections, we hypothesize the following: additional furniture introduces many absorbing corners and surfaces that promote faster energy decay, which reduces the effective late reverberation and makes T60 easier to learn in the furnished setting. At the same time, the irregular furniture layout creates more complex early multiple reflections, which makes EDT and C50 slightly harder to match in the furnished configuration.

---

> ### Author Response · Authors · 2025-11-26
> **Question 4**
>
> Thank you for raising the question regarding our decision to predict only the magnitude of the STFT.
> We provide clarification below on why this choice is aligned with prior work and remains practically effective.
>
> First, in the official implementation of Neural Acoustic Fields (NAF), the authors report that predicting only the magnitude of the STFT, followed by Griffin--Lim reconstruction, achieved comparable or better results relative to variants that attempted to include phase. This behavior is explicitly documented in the NAF GitHub repository and reflects their empirical observations.
>
> Second, recent works most relevant to our task, such as AV-NeRF and NeRAF, also adopt magnitude-only STFT prediction and reconstruct the waveform using Griffin--Lim. These methods do not regress phase, largely because phase varies rapidly with small spatial shifts and is sensitive to pose noise or discretization, while magnitude-only prediction has shown stable and competitive performance in practice. Our design follows the same strategy used in these recent neural acoustic synthesis approaches.
>
> In summary, prior work (NAF) has observed that magnitude-only prediction achieves comparable or better performance than phase-inclusive variants, and recent models relevant to our setting (AV-NeRF, NeRAF) adopt the same magnitude-only strategy. For these reasons, we believe that this design choice is well supported by existing practice and appropriate for our problem formulation.

---

### Author Response · Authors · 2025-11-29
**Overview of Revisions and Clarifications**

We sincerely thank the reviewers for their constructive feedback. Below we summarize the key clarifications and improvements made during the rebuttal:

- **Physics-informed wave propagation priors**
  We clarified that the pressure map features inherently capture diffraction and multiple reflections through physics-governed update rules in the FDTD solver, providing strong inductive bias even with simplified simulation settings.

- **Robustness to noisy geometry**
  We highlighted existing analysis in **Appendix D**, showing that WavNAF consistently benefits from imperfect geometry and acoustic parameter estimation, confirming that accurate absolute material properties are not required.

- **Improved visual evidence**
  We added clearer visualizations of simulated pressure fields and predicted loudness maps in **Appendix E** and the supplementary material, addressing qualitative concerns.

- **Data efficiency**
  Results under reduced supervision show that WavNAF trained with **50%** of RAF data achieves **better EDT** than NeRAF trained with **75%**, confirming reliable early-reflection recovery with substantially less data.

- **Strengthened SOTA performance across all RAF metrics**
  With improved initialization, WavNAF achieves enhanced performance in **T60**, **C50**, **EDT**, and **STFT error** on RAF:

  | Metric | NeRAF  | WavNAF  | Improvement |
  |--------|---------|---------|-------------|
  | T60 ↓   | 7.47    | **7.36** | ✓ Better late reverberation modeling |
  | C50 ↓   | 0.61    | **0.59** | ✓ Improved clarity |
  | EDT ↓   | 0.020   | **0.018** | ✓ Strong early-reflection recovery |
  | STFT error ↓ | 0.17 | **0.16** | ✓ Reduced spectral reconstruction error |

**Conclusion**
These clarifications and updates further reinforce our core claim: physics-informed wave priors significantly improve neural acoustic field modeling, resulting in strong accuracy, robustness, and efficiency across diverse scenarios.

---

### Meta-Review · Area_Chair_rM7F · 2025-12-21

**Summary:**

The recommendation of rejection is mainly based on the fact that several important reviewers' concerns are still not well addressed, detailed below. The AC thinks the paper studies an interesting problem, and suggests the authors to take reviewers' suggestions into concern and update the paper for resubmission.

1. In the rebuttal, the authors replaced their original MLP initialization with Kaiming initialization and evaluated WavNAF on the RAF dataset, leading to large performance gains. Due to the limited rebuttal time, they could not rerun all experiments in the paper with the updated initialization. This is one of my main reasons for rejection. The AC believes this is a very significant change that has to be made. Rerunning all experiments with the new initialization will update all numbers in the original submission, which should be addressed in a resubmission rather than updating in camera ready.

2. The wave propagation simulation is very simplified, but used as the main novelty that leads to the performance gains. According to the authors, "FDTD stage is deliberately simplified and scaled down; 2D floor-plane grid in the normalized NeRF space, which is much coarser than the true 3D room and ignores vertical propagation and ceiling reflections; just used as wave propagation prior". The reviewers are questioning about how this simplified simulation setup can help, and more analysis and explanations are needed.

3. The concerns of "lack of visualizations" are mostly addressed, though no qualitative audio examples are provided to get a better sense of the spatial audio rendering quality.

4. The reviewers also suggest many missing references that are very relevant to the work, which the authors should add, discuss, or even compare in Related Work in resubmission.

**Reviewer Concerns:**

***Reviewers' concerns that were addressed by the rebuttal:***

Questions on Time cost, sound speed, missing baselines, specification of some components, visualizations

***Reviewers' concerns that are still outstanding:***

how vision helps, 2D simulation not as realistic, initialization strategies on performance, etc.

See details below

**Reviewer Scores:**

***Reviewer QtCm is likely to maintain the score of 4, as the major concerns (1 and 2) are not well addressed in the rebuttal.***

1. lack of clear verification to show the necessity of using vision to assist spatial audio propagation process

Not well addressed

2. vision alone can't fully capture all of these influencing factors, but the authors claim modeling factors like diffraction, diffusion and reflection without much explanation

Not well addressed

3. Lack of visualization

Mostly addressed

4. Why ignoring phase and just predicting STFT map?

Partially addressed

5. Some minor questions about sound speed and missing baselines

Addressed

***Reviewer k2CB is also likely to maintain his original rating of 4 as some main concerns (1 and 2 below) are not well addressed.***

1. 2-D pressure field and wave simulation not realistic.

Not well addressed

2. Modest performance improvement on RAF

Partially addressed

3. Missing qualitative visualizations

Mostly addressed

4. Why not directly using the pressure map simulation that is already estimated?

Mostly addressed

5. Time cost

Addressed

6. Why predicting STFT rather than impulse response directly?

Partially addressed

7. Missing references of latest papers

Promised to address


***Reviewer 3MYx is likely to maintain the original score of 8 as this reviewer is most positive initially and the questions are mostly addressed. However, his reviewer could also lower the score given the shared concerns raised by other reviewers.***

1. different initialization strategies for the MLPs

Partially addressed

2. FDTD simulations significantly increases the computational complexity

Mostly addressed

3. data efficiency hold true when testing on the RAF real-world dataset?

Mostly addressed

***Review 44Hp is likely to maintain the score of 4 as some questions are fundamentally hard to address with a rebuttal like 2 and 3 below***

1. Some components are underspecified

Mostly addressed

2. The method relies heavily on empirical choices like the mapping from optical opacity to acoustic parameters, but lacks thorough ablations to justify their ranges and robustness.

Partially addressed

3. only considers a 2D structure when synthesizing acoustic signals

Partially addressed

---

### Decision · Program_Chairs · 2026-01-26

Reject